# An unexpected INAD PDZ tandem-mediated plcβ binding in *Drosophila* photo receptors

**Fei Ye[1,2†], Yuxin Huang[3†], Jianchao Li[1], Yuqian Ma[4], Chensu Xie[1], Zexu Liu[1], Xiaoying Deng[3], Jun Wan[1,3], Tian Xue[4], Wei Liu[3]\*, Mingjie Zhang[1,3]\***

[1]Division of Life Science, State Key Laboratory of Molecular Neuroscience, Hong Kong University of Science and Technology, Hong Kong, China; [2]Institute for Advanced Study, Hong Kong University of Science and Technology, Hong Kong, China; [3]Shenzhen Key Laboratory for Neuronal Structural Biology, Biomedical Research Institute, Shenzhen Peking University-The Hong Kong University of Science and Technology Medical Center, Shenzhen, China; [4]Hefei National Laboratory for Physical Sciences at Microscale,CAS Key Laboratory of Brain Function and Disease, Neurodegenerative Disorder Research Center, School of Life Sciences, University of Science and Technology of China, Hefei, China

**Abstract** INAD assembles key enzymes of the *Drosophila* compound eye photo-transduction pathway into a supramolecular complex, supporting efficient and fast light signaling. However, the molecular mechanism that governs the interaction between INAD and NORPA (phospholipase Cβ, PLCβ), a key step for the fast kinetics of the light signaling, is not known. Here, we show that the NORPA C-terminal coiled-coil domain and PDZ-binding motif (CC-PBM) synergistically bind to INAD PDZ45 tandem with an unexpected mode and unprecedented high affinity. Guided by the structure of the INAD–NORPA complex, we discover that INADL is probably a mammalian counterpart of INAD. The INADL PDZ89 tandem specifically binds to PLCβ4 with a mode that is strikingly similar to that of the INAD–NORPA complex, as revealed by the structure of the INADL PDZ89–PLCβ4 CC-PBM complex. Therefore, our study suggests that the highly specific PDZ tandem – PLCβ interactions are an evolutionarily conserved mechanism in PLCβ signaling in the animal kingdom.
DOI: https://doi.org/10.7554/eLife.41848.001

**\*For correspondence:**
liuwei@sphmc.org (WL);
mzhang@ust.hk (MZ)

[†]These authors contributed equally to this work

## Introduction

Scaffold proteins can serve as platforms for the assembly of signaling components into macromolecular complexes, targeting them to specific cellular localizations, as well as actively modulating signaling processes (*Bhattacharyya et al., 2006*; *Pawson and Nash, 2003*; *Zhang and Wang, 2003*). They therefore support the occurrence of signaling events in precise locations and at specific time points in different tissues. The *Drosophila* compound eye rhodopsin-mediated photo-transduction signaling process is highly elaborate. It is also one of the best-studied model systems showing how light signals, via INAD scaffold-organized signaling complexes, can be transduced at a very large dynamic range with extremely rapid kinetics and intricate regulatory mechanisms (*Huber, 2001*; *Li and Montell, 2000*; *Liu et al., 2011*; *Mishra et al., 2007*; *Tsunoda et al., 1997*; *Tsunoda and Zuker, 1999*).

At the inner surface of fly photoreceptor rhabdomeric membranes, the master scaffold protein INAD (encoded by *inaD* for inactivation no after potential D and contains 5 PDZ domains arranged in tandem; *Figure 1A*) forms stoichiometric multi-molecular complexes with phospholipase Cβ

**Figure 1.** Super strong interaction between NORPA and INAD. (**A**) Schematic cartoon diagram showing the pathway of *Drosophila* photo-transduction signaling. (**B**) Schematic diagram showing the domain organizations of NORPA and INAD. The interaction mediated by NORPA CC-PBM and INAD PDZ45 is illustrated. The color coding of the domains is kept throughout this paper. (**C**) Isothermal titration calorimetry (ITC)-based measurement of the binding between NORPA CC-PBM and INAD PDZ45 (**C1**), and between the 8KA mutation of NORPA CC-PBM and INAD PDZ45 (**C2**). The sites of the point mutations in the CC region of NORPA are indicated by a green dot. (**D**) Table summarizing the measured binding affinities between various forms of NORPA CC-PBM and INAD derived from ITC-based assays.

DOI: https://doi.org/10.7554/eLife.41848.002

The following figure supplement is available for figure 1:

**Figure supplement 1.** Characterization of the interaction between INAD PDZ45 and NORPA CC-PBM.

DOI: https://doi.org/10.7554/eLife.41848.003

(NORPA), $Ca^{2+}$-permeable transient receptor potential (TRP) channel and eye-specific protein kinase C (eye-PKC) (*Adamski et al., 1998a*; *Chevesich et al., 1997*; *Huber et al., 1996*; *Kimple et al., 2001*; *Liu et al., 2007*; *Montell, 2005*; *Peng et al., 2008*; *Shieh and Zhu, 1996*; *Tsunoda et al., 1997*; *van Huizen et al., 1998*; *Ye et al., 2016*). Genetic, cell biology, biochemistry, and structural biology studies have revealed that INAD PDZ2 is required for binding to eye-PKC, INAD PDZ3 for interaction with TRP channel, and INAD PDZ5 for engaging NORPA (*Adamski et al., 1998b*; *Chevesich et al., 1997*; *Tsunoda et al., 1997*; *Ye et al., 2016*). Mutations of *inaD* that lead to disruption of each of these interactions invariably impair fly photo signal transduction (*Scott and Zuker, 1998*). Fly photoreceptorsare capable of responding, via the INAD-mediated assembly of the signaling complex, to light signals with extremely fast response time and termination kinetics (*Henderson et al., 2000*; *Ranganathan et al., 1995*). One critical step in fast light signaling in fly photoreceptors is the efficient coupling of NORPA, one of the fastest enzymes catalyzing PtdIns(4,5) $P_2$ hydrolysis, to the other INAD-organized signaling components (*Minke and Parnas, 2006*; *Shieh et al., 1997*). Previous genetic studies have revealed that a mutation within PDZ5 of INAD (*inaD²*) or a single point mutation within the PBM of NORPA could selectively impair the INAD–NORPA interaction, indicating that the fifth PDZ domain of INAD and the PBM of NORPA are required for INAD–NORPA interaction (*Cook et al., 2000*; *Shieh et al., 1997*; *Tsunoda et al.,*

*1997*). The direct interaction between NORPA and PDZ5 is, however, extremely weak ($K_d$ ~560 μM) (*Liu et al., 2011*) and is unlikely to be capable of supporting the specific interaction between INAD and NORPA. It is possible that the INAD–NORPA interaction follows a mode that is entirely different from all known PDZ domain-mediated target-recognition modes.

In this study, we show that the PDZ45 tandem of INAD functions as a supramodule binding to the entire C-terminal coiled-coil domain and PDZ-binding motif of NORPA (CC-PBM) with an unexpectedly high affinity. The crystal structure of INAD PDZ45 in complex with NORPA CC-PBM uncovers a highly unusual PDZ domain – target binding mode and explains the high binding affinity and specificity between INAD and NORPA. Guided by the INAD–NORPA complex structure, we discover that in the vertebrate system, INADL but not MUPP1, a close paralogue of INADL, specifically binds to PLCβ4. Our biochemical and structural studies demonstrate that INADL PDZ89 forms a supramodule and binds to the entire C-terminal coiled-coil domain of PLCβ4 in a manner that is strikingly similar to that of the INAD–NORPA complex. The striking similarity between the PDZ scaffold and PLCβ interactions at the molecular level might point to an evolutionarily conserved molecular adaption in PLCβ signaling in the animal kingdom.

## Results

### NORPA CC-PBM synergistically interacts with the INAD PDZ45 tandem with a very high affinity

Before performing detailed biochemical characterization of the INAD–NORPA interaction, we carefully analyzed the NORPA protein sequence and found that the C-terminal region of NORPA contains a coiled coil (CC) (residues 850–1086) immediately followed by a PDZ binding motif (PBM) (residues 1092–1095, amino acid 'EFYA') (*Figure 1B*). The predicted coiled-coil and PBM are separated by just six amino acid residues, suggesting the possibility that the coiled coil may cooperate with the PBM for NORPA to bind to INAD. To test this hypothesis, we characterized the INAD–NORPA interaction quantitatively using purified proteins (*Figure 1C and D*). Isothermal titration calorimetry (ITC) analysis revealed that the full-length INAD binds to NORPA CC-PBM with a $K_d$ ~8 nM (*Figure 1D* and *Figure 1—figure supplement 1B*), an affinity that is much stronger than those of all known canonical PDZ–target interactions (usually with $K_d$ of a few to a few tens of μMs) (*Ye and Zhang, 2013*). Further mapping showed that INAD PDZ45 is the minimal and complete region of INAD in binding to NORPA CC-PBM, as the PDZ45 tandem binds to NORPA CC-PBM with an affinity similar to that of the full-length INAD (*Figure 1C1 and D*). Deletion of the coiled-coil domain from the NORPA CC-PBM decreased its binding to INAD PDZ45 by ~2400 fold, and removal of the three-residue PBM completely eliminated NORPA CC-PBM's binding to INAD PDZ45 (*Figure 1D* and *Figure 1—figure supplement 1A and C*), indicating that the coiled coil and PBM of NORPA function synergistically in binding to INAD PDZ45. We showed earlier that PDZ45 forms a supramodule that is necessary for fly visual signaling (*Liu et al., 2011*). Consistent with this finding, isolated PDZ5 displayed a weak binding to NORPA CC-PBM (*Figure 1D* and *Figure 1—figure supplement 1D*). Taken together, the above biochemical results revealed that the INAD PDZ45 tandem binds to NORPA CC-PBM with a very high affinity. It is noted that the binding affinity between INAD PDZ45 and NORPA CC-PBM is among the tightest in all known PDZ–target interactions (*Ye and Zhang, 2013*), and that the involvement of a coiled-coil structure in binding to PDZ domains is highly unusual and previously unknown.

### Structural characterization of the NORPA–INAD complex

Having characterized the detailed interaction between NOROA and INAD, we wanted to understand the molecular mechanism governing this specific binding by trying to solve the atomic structure of the INAD PDZ45 and NORPA CC-PBM complex using X-ray crystallography. Multiple years of attempts to crystallize the complex prepared from the wildtype INAD PDZ45 and NORPA CC-PBM all failed, even though our NMR-based study indicated that the PDZ45–NORPA CC-PBM complex is conformationally homogeneous in solution (*Figure 2—figure supplement 1A*). Confronted by this disappointing result, we decided to take a step back and tried to solve the NORPA CC-PBM structure alone first, hoping to be able to get some clues that might lead us to a way to obtain PDZ45–

NORPA CC-PBM complex crystals (the structure of INAD PDZ45 was solved in our earlier study (*Liu et al., 2011*)).

Using lysine reductive-methylated protein, we were able to solve the structure of NORPA CC-PBM at the resolution of 3.0 Å (*Supplementary file 1* and *Figure 2—figure supplement 2A*). Like the coiled-coil structure of mammalian PLCβ3 (*Lyon et al., 2013*), NORPA CC folds into an elongated three helix bundle structure with an N-terminal short helix coupling to the middle region of the helix bundle (*Figure 2—figure supplement 2A*). The C-terminal 16 residues (aa 1080–1095) that include the PBM is disordered and invisible in the structure (*Figure 2—figure supplement 3C*). The packing surface of NORPA CC-PBM crystals involves an extensive surface composed of α1–α4 (*Figure 2—figure supplement 2C*). We reasoned that this packing surface might overlap with the INAD PDZ45 binding surface, and therefore prevented the same surface from being used for crystal packing in the PDZ45–NORPA CC-PBM complex (also see *Figure 2—figure supplement 2D*). We noted that the solvent-exposed face of the α2 helix is lined with eight Lys residues, which are all evolutionarily conserved and required for the membrane binding of NORPA CC (*Figure 2—figure supplement 2B*, and see below for more details) (*Lyon et al., 2013*). We substituted the eight Lys residues with Ala residues (designated as CC$^{8KA}$-PBM) with the rationale that such substitutions should not impair the helical conformation of α2 but might create a new crystal packing surface for the PDZ45–NORPA CC-PBM complex (*Figure 2—figure supplement 2D*). A circular dichroism spectroscopic study showed that the Ala substitutions did not introduce obvious conformational perturbations to NORPA CC-PBM (*Figure 2—figure supplement 2E*). Importantly, the NORPA CC$^{8KA}$-PBM mutant binds to INAD PDZ45 with an affinity comparable to that of the wildtype NORPA CC-PBM (*Figure 1C2 and D*). Satisfyingly, the NORPA CC$^{8KA}$-PBM–INAD PDZ45 complex could be crystallized and the structure of the complex was solved at the resolution of 3.2 Å by molecular replacement methods, using the apo-form NORPA CC-PBM structure as the search model (*Supplementary file 1* and *Figure 2—figure supplement 3D, E*). Matching our initial design, the Ala-containing α2 helix was indeed the major part of the packing surface for the NORPA CC$^{8KA}$-PBM–INAD PDZ45 complex crystals (*Figure 2—figure supplement 2*D).

The most important feature of the complex is that both PDZ domains of the PDZ45 tandem and both the coiled coil and PBM of NORPA participate cooperatively to form a highly stable INAD–NORPA complex burying an extensive interaction surface area of ~1373 Å$^2$ (*Figure 2A*). The PDZ45–NORPA CC-PBM complex structure also reveals a hitherto uncharacterized interaction mode for PDZ domains. PDZ45 binding stabilizes the conformation of NORPA's C-terminal 16 amino acids by extending the α4 helix for five more turns, immediately followed by a four-residue β-strand corresponding to the PBM of NORPA (*Figure 2B* and *Figure 2—figure supplement 3C*). The conformational rigidity between the α4 helix and the PBM in the PDZ45–NORPA CC-PBM complex shows that the CC and PBM of NORPA function synergistically in binding to INAD PDZ45. Consistent with this notion, the insertion of four flexible residues ('GSGS') between the CC and PBM of NORPA weakened its binding to INAD PDZ45 by ~110 fold (*Figure 2H* and *Figure 2—figure supplement 4A5*). Structural alignment analysis showed that the INAD PDZ45 tandem, including the inter-PDZ domain interactions, underwent minimal conformational changes upon binding to NORPA CC-PBM (*Figure 2C*) (*Liu et al., 2011*).

The binding interface of the PDZ45–NORPA CC-PBM complex can be divided into three distinct sites (*Figure 2D*): the PDZ5–PBM interaction site (site 1, *Figure 2E*), the PDZ5/CC packing site (site 2, *Figure 2F*), and the PDZ4/CC binding site (site 3, *Figure 2G*). In site 1, NORPA PBM ('EFYA') inserts into the βB'/αB' groove of PDZ5 following the typical PDZ–ligand interaction mode (*Figure 2B and E*) (*Ye and Zhang, 2013*). We note that the last residue of the NORPA PBM is Ala, but Ala at the 0 position is very rare for canonical PBMs (*Ye and Zhang, 2013*). Structural analysis revealed that the pocket of PDZ5 that accommodates the sidechain of the 0 position residue of the NORPA PBM is shallow and enriched in bulky aromatic residues (F632, F649, F642 and F-2, *Figure 2E* and *Figure 2—figure supplement 5A*), which does not favor hydrophobic residues with bulky sidechains. Indeed, substitution of A0 with Leu of the NORPA CC-PBM led to a ~4-fold decrease in the affinity of binding to INAD PDZ45 (*Figure 2H* and *Figure 2—figure supplement 4A*). F-2 from the NORPA PBM extensively interacts with a host of hydrophobic residues from PDZ5 (*Figure 2E*). Substitution of F-2 with Ala completely eliminated the binding of the NORPA CC-PBM to INAD PDZ45 (*Figure 2H* and *Figure 2—figure supplement 4A*). Site 2 accomodates extensive interactions between the NORPA CC-PBM and PDZ45 outside the PBM binding pocket of PDZ5.

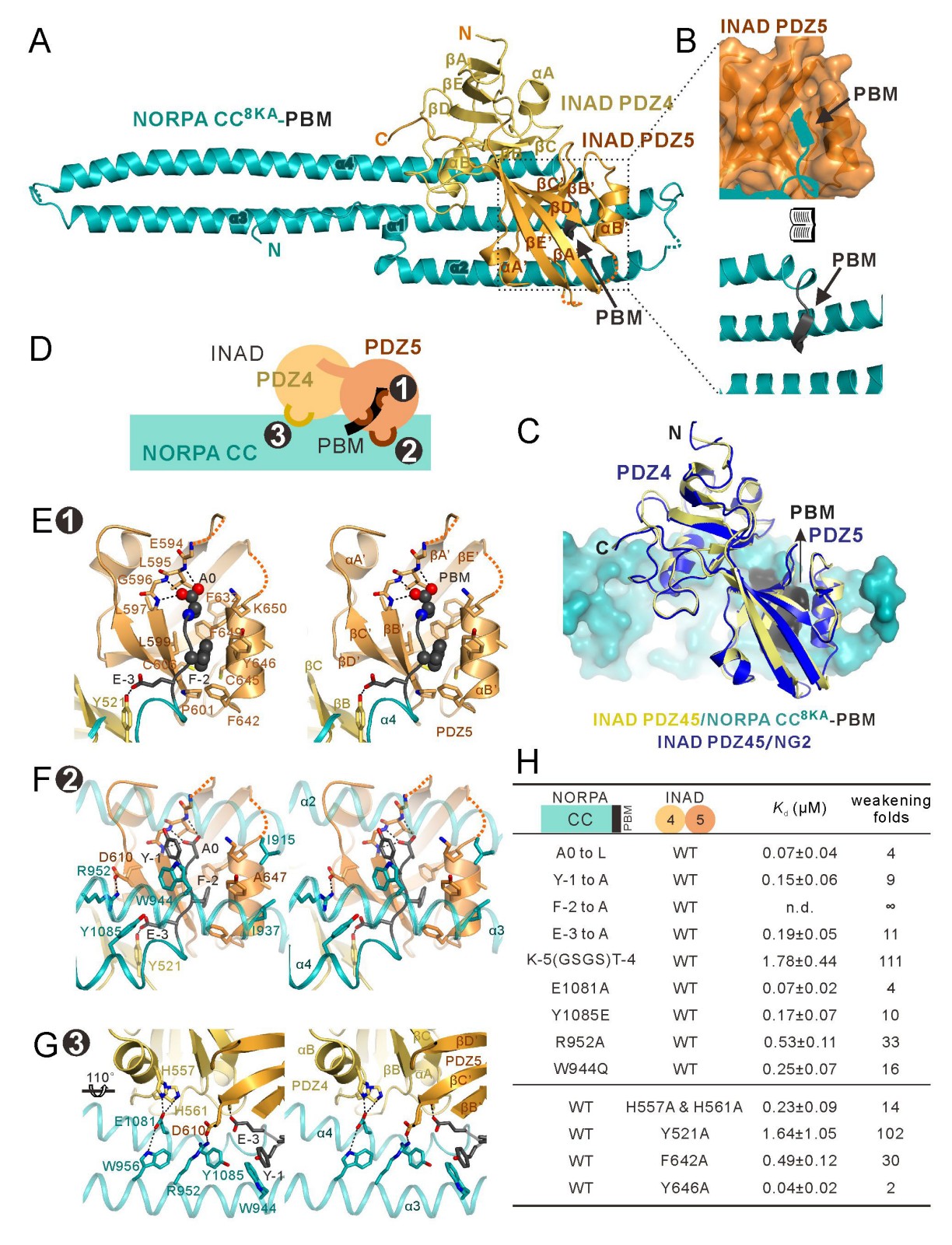

| NORPA | | INAD | | $K_d$ (μM) | weakening folds |
|---|---|---|---|---|---|
| CC | PBM | 4 | 5 | | |
| A0 to L | | WT | | 0.07±0.04 | 4 |
| Y-1 to A | | WT | | 0.15±0.06 | 9 |
| F-2 to A | | WT | | n.d. | ∞ |
| E-3 to A | | WT | | 0.19±0.05 | 11 |
| K-5(GSGS)T-4 | | WT | | 1.78±0.44 | 111 |
| E1081A | | WT | | 0.07±0.02 | 4 |
| Y1085E | | WT | | 0.17±0.07 | 10 |
| R952A | | WT | | 0.53±0.11 | 33 |
| W944Q | | WT | | 0.25±0.07 | 16 |
| WT | | H557A & H561A | | 0.23±0.09 | 14 |
| WT | | Y521A | | 1.64±1.05 | 102 |
| WT | | F642A | | 0.49±0.12 | 30 |
| WT | | Y646A | | 0.04±0.02 | 2 |

**Figure 2.** Structure of the NORPA CC^8KA-PBM–INAD PDZ45 complex. (**A**) Ribbon representation of the NORPA CC^8KA-PBM–INAD PDZ45 complex structure. The disordered loops are drawn as dashed lines in the ribbon representation. (**B**) Open book view showing the binding interface between INAD PDZ5 and NORPA PBM. (**C**) Superimposition of INAD PDZ45–NG2 peptide structure (blue) over PDZ45 in complex with NORPA (yellow). (**D**) Schematic cartoon diagram summarizing the binding mode of the NORPA CC^8KA-PBM–INAD PDZ45 complex with three characteristic binding sites

*Figure 2 continued on next page*

*Figure 2 continued*

detailed in panels E–G. (E–G) Stereoviews showing the interaction interfaces between INAD PDZ45 and NORPA CC-PBM. The side chains of the residues involved in the inter-domain interactions are drawn in the stick representation. The complex interface is divided into three parts, the PDZ5–PBM interaction site ((E) site 1), the PDZ5/CC packing site ((F) site 2), and the PDZ4/CC binding site ((G) site 3). (H) Table summarizing the measured binding affinities, showing that mutations of the critical residues in the NORPA CC-PBM–INAD PDZ45 interface weakened or even abolished the interaction.

DOI: https://doi.org/10.7554/eLife.41848.004

The following figure supplements are available for figure 2:

**Figure supplement 1.** NMR-spectroscopy-based characterization of the INAD PDZ45–NORPA CC-PBM interaction.

DOI: https://doi.org/10.7554/eLife.41848.005

**Figure supplement 2.** Crystal structure of the NORPA CC-PBM domain.

DOI: https://doi.org/10.7554/eLife.41848.006

**Figure supplement 3.** Alignment of the apo NORPA CC-PBM and the NORPA CC[8KA]-PBM–INAD PDZ45 complex structures showing the INAD PDZ45 binding-induced conformational changes of the NORPA CC-PBM.

DOI: https://doi.org/10.7554/eLife.41848.007

**Figure supplement 4.** Impact of the mutations on the binding affinity between INAD PDZ45 and the NORPA CC-PBM.

DOI: https://doi.org/10.7554/eLife.41848.008

**Figure supplement 5.** Biochemical and structural analysis of the INAD PDZ45–NORPA CC-PBM interaction.

DOI: https://doi.org/10.7554/eLife.41848.009

For example, Y-1 from the NORPA PBM interacts with W944 from α3 of the NORPA CC domain; E-3 from the PBM forms hydrogen bonds with Y521 from βB of PDZ4 and with Y1085 from α4 of the NORPA CC; R952 from α3 of NORPA forms salt bridges with D610 from βC′ of PDZ5 (*Figure 2F*). In site 3, two His residues from INAD PDZ4 form hydrogen bonds with E1081 from α4 of the NORPA CC, and these residues are further stabilized by a hydrogen bond with W956 from α3 of the NORPA CC (*Figure 2G*). Mutations of individual residues in these binding sites invariably weakened the NORPA–INAD interaction (*Figure 2H* and *Figure 2—figure supplement 4*). We further used NMR spectroscopy to investigate the binding of the NORPA CC-PBM to INAD PDZ45 in solution. Comparison of the $^1$H-$^{15}$N heteronuclear single quantum coherence spectroscopy (HSQC) spectrum of $^{15}$N-labeled INAD PDZ45 alone with that of $^{15}$N-labeled PDZ45 in complex with an unlabeled NORPA CC-PBM showed that the regions of PDZ45 that underwent NORPA-binding-induced chemical shift changes were clustered in the three sites analyzed above (*Figure 2—figure supplement 1B*), further confirming the interaction between the NORPA CC-PBM and PDZ45 observed in the crystal structure.

The NORPA CC-PBM–INAD PDZ45 complex structure provides a clear mechanistic explanation of why the *inaD*[2] mutant (a missense mutation changing Gly605 in PDZ5 to Glu) led to mislocalization of NORPA in photoreceptor cells and to severe defects in photo-response amplitude and kinetics (*Cook et al., 2000*; *Tsunoda et al., 1997*). G605 is in a hydrophobic core that couples PDZ45 into a structural supramodule (*Figure 2—figure supplement 5B*). The G605E mutation would introduce a negatively charged sidechain to this hydrophobic core, and thus is expected to destabilize the PDZ45 supramodule and impair its binding to NORPA. Consistent with this analysis, the PDZ45[G605E] protein showed significantly decreased binding to NORPA (*Figure 2—figure supplement 5C*), showing that the structural integrity of the INAD PDZ45 supramodule is important for photo signaling in *Drosophila* photopreceptors. To further confirm this, we generated a T669E mutant of PDZ45, which was previously found to impair conformational coupling between PDZ4 and PDZ5 (*Liu et al., 2011*). Pulldown analysis showed that the T669E mutant of INAD PDZ45 had significantly weaker binding to NORPA CC-PBM than wildtype INAD PDZ45 (*Figure 2—figure supplement 5D*), further indicating that the formation of the PDZ45 supramodule is important for the strong binding of INAD to NORPA.

## INAD-mediated membrane micro-domain organization of the fly photosignal transduction complex

A number of studies have suggested that the PLCβ CC domain is important for the membrane targeting of the enzyme, an event that is necessary for the full activity of the enzyme in vivo (*Lee et al., 1993*; *Lyon et al., 2013*; *Park et al., 1993*; *Waldo et al., 2010*). We next analyzed whether the

formation of the tight complex between INAD PDZ45 and the NORPA CC-PBM might influence the membrane binding of the NORPA CC. The evolutionarily conserved, positively charged residues in the α2 helix of the NORPA CC are aligned on a flat and exposed surface of the domain (*Figure 3A* and *Figure 2—figure supplement 2B*). A liposome sedimentation-based assay showed that the NORPA CC-PBM strongly interacted with lipid membranes (*Figure 3B1 and C*). By contrast, the NORPA CC8KA-PBM showed significantly decreased membrane-binding capacity, indicating that the positively charged α2 helix is essential for membrane binding (*Figure 3B2 and C*). In the NORPA–INAD complex, INAD PDZ45 binds to one face of the NORPA CC with the two PDZ domains crossing over the α2, α3, and α4 helices (*Figure 2A*). Interestingly, one end of PDZ5 (formed by the βA/βB-loop, the βC/αA-loop, and the αB/βF-loop) is positively charged and aligned with the NORPA CC α2 helix forming a flat surface (*Figure 3A*), indicating that binding to INAD PDZ45 does not interfere with the membrane binding of the NORPA CC-PBM mediated by its α2 helix. Consistent with this structural analysis, the INAD PDZ45–NORPA CC-PBM complex displayed comparable (or somewhat enhanced) lipid membrane binding when compared with the NORPA CC-PBM alone (*Figure 3B3 and C*). Our previous study showed that INAD PDZ3 tethers the TRP channel through specific and strong binding to the TRP channel C-terminal tail (*Ye et al., 2016*). Therefore, the INAD PDZ345 tandem can simultaneously recruit NORPA and the TRP channel, forming a closely

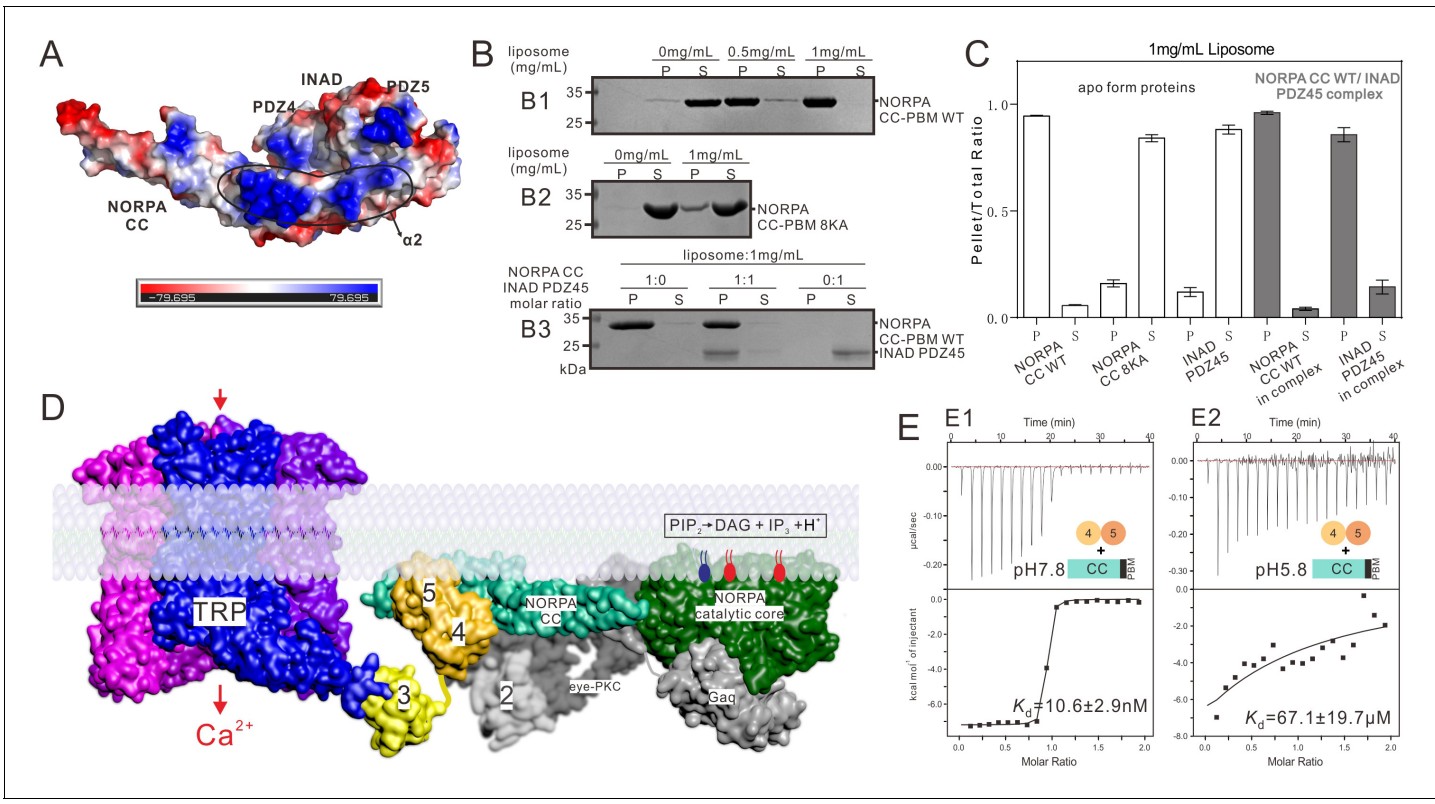

**Figure 3.** Summary model of the INAD PDZ45–NORPA CC-PBM interaction in *Drosophila* photon-transduction. (**A**) Surface representation showing the electrostatic potential of the INAD PDZ45–NORPA CC-PBM complex. The ± 80 kT/e potential isocontours are shown as blue (positively charged) and red (negatively charged) surfaces, respectively. This electrostatic potential analysis was generated by Pymol (https://www.pymol.org). (**B**) Lipid sedimentation assay showing the binding properties of NORPA CC-PBM WT (**B1**), NORPA CC8KA-PBM mutant (**B2**), and the INAD PDZ45–NORPA CC-PBM complex (**B3**) to liposomes prepared from bovine brain lipid extracts. Fractions labeled 'S' and 'P' represent proteins that are present in the supernatants and pellets after centrifugation, denoting lipid-free and lipid-bound forms of the proteins, respectively. (**C**) Quantification of the sedimentation-based assay of the lipid binding capacities of the proteins shown in panel B. The results are from three independent batches of sedimentation assays and are represented as mean ± SD. (**D**) Surface combined cartoon representation showing a model of the INAD-organized signaling complex underneath the rhabdomere plasma membranes. In this model, the INAD PDZ345 tandem can position NORPA and the TRP channel in close proximity on the 2D membrane plane. (**E**) ITC-based measurement of the binding between the NORPA CC-PBM and INAD PDZ45 at pH 7.8 (**E1**) and at pH 5.8 (**E2**), showing acidification-induced weakening of the binding between NORPA and INAD.
DOI: https://doi.org/10.7554/eLife.41848.010

arranged signaling membrane micro-domain (*Figure 3D*). It is envisioned that the physical positioning of NORPA right next to the TRP channel allows the channel to sense efficiently the membrane lipid component changes initiated by light-induced phosphatidylinositol 4,5-bisphosphate ($PIP_2$) hydrolysis by NORPA. It is noted that eye protein kinase C (eye-PKC) is also recruited to this membrane micro-domain through binding to the PDZ2 domain of INAD (*Figure 3D*).

An important question is how the nano-molar strong interaction between INAD and NORPA is regulated in vivo during the deactivation of the photo-transduction process. Previous studies showed that light-induced acidification that results from the NORPA-mediated hydrolysis of $PIP_2$ can lead to conformational uncoupling of the INAD PDZ45 tandem (*Huang et al., 2010*; *Liu et al., 2011*). Interestingly, we found that decreasing the pH of the buffer from 7.8 to ~5.8 dramatically weakened the binding between INAD and NORPA by ~6700-fold (*Figure 3E*), supporting the notion that the light-induced acidification of the rhabdomeric compartment could dissociate the INAD–NORPA complex.

## PLCβ4 binding to INADL and NORPA binding to INAD occur through similar modes

Next, we wanted to test whether the unexpected INAD PDZ tandem-mediated NORPA binding mode could also occur for the mammalian PLCβ enzymes. No mammalian homologue of INAD has been identified, although mammals do contain several multi-PDZ-domain proteins including one called INAD-like (INADL, also known as PATJ), which contains a total of 10 PDZ domains. A previous study using microarray-based technique found that, upon exposure to light, the expression of the *INADL* and *Slc9a3r1* (encoding NHERF1) genes were upregulated in mice lacking both rods and cones (*Peirson et al., 2007*). Recent human genetic studies of sleeping-disorder patients showed that *INADL* might be associated with multiple sleeping disorders and circadian timing variations (*Forni et al., 2014*; *Jones et al., 2016*; *Lane et al., 2017*). Considering that melanopsin is known to be a sleep modulator, and that melanopsin polymorphisms have been associated with circadian dysfunction (*Lee et al., 2014*; *Roecklein et al., 2012*), we hypothesized that INADL might function as a scaffold protein in the melanopsin-mediated intrinsically photosensitive retinal ganglion cells (ipRGC) in mammals.

PLCβ4, a vertebrate orthologue of NORPA in *Drosophila*, was previously found to be a key regulator of melanopsin signaling in ipRGC (*Xue et al., 2011*). Amino-acid sequence analysis revealed that the CC-PBM domains of NORPA and PLCβ4 share a sequence identity of 27% (*Figure 4A*). More importantly, PLCβ4 shares the following features with the NORPA CC-PBM that are important for both INAD PDZ45 and lipid membrane binding: (i) PLCβ4 also contains eight Lys residues in the predicted α2 helix with positions identical to those in NORPA (highlighted with green dots in *Figure 4A*); (ii) PLCβ4 also contains a highly conserved PBM with the sequence 'ATVV' (highlighted with blue dots in *Figure 4A*), the PLCβ4 PBM also immediately follows the α4 of the predicted coiled-coil domain (*Figure 4A and B*), indicating that the coiled coil and PBM of PLCβ4 may also function synergistically in binding to PDZ scaffold proteins; and (iii) a number of residues from NORPA CC domains that directly interact with INAD PDZ45 also exist in the same positions in the PLCβ4 CC (highlighted with black dots in *Figure 4A*). Therefore, we hypothesized that PLCβ4 may also bind to certain PDZ scaffold protein(s) in the same way that NORPA does to INAD.

To test this hypothesis, we screened for possible PDZ proteins that may interact with the PLCβ4 CC-PBM with high affinity using quantitative ITC-based binding assay coupled with size exclusion chromatography using purified proteins. We focused our efforts on multi-PDZ mammalian scaffold proteins that either share certain similarity to INAD (e.g., have five or more PDZ domains) or have been indicated to be related to visual signaling. This narrowed our targets to the following four proteins: INADL, MUPP1, GRIP1, and NHERF. We used multiple PDZ domains from these proteins (i.e. two or more consecutive PDZ domains) to test their possible binding to the PLCβ4 CC-PBM (*Figure 4—figure supplement 1*). Among the numerous PDZ tandems tested, we found that only the INADL PDZ89 tandem interacted with the PLCβ4 CC-PBM with a strong affinity (Kd ~0.2 μM, *Figure 4B–D*). The other purified proteins either did not bind or bound with very low affinities to the PLCβ4 CC-PBM (*Figure 4—figure supplement 1*), indicating that the INADL PDZ89 and PLCβ4 interaction is very specific. Further mapping of the interaction revealed that, as in the NORPA CC-PBM–INAD PDZ45 interaction, both the CC and the PBM of PLCβ4 are required for its binding to INADL PDZ89 (*Figure 4D* and *Figure 4—figure supplement 2*). Conversely, isolated PDZ8 or PDZ9

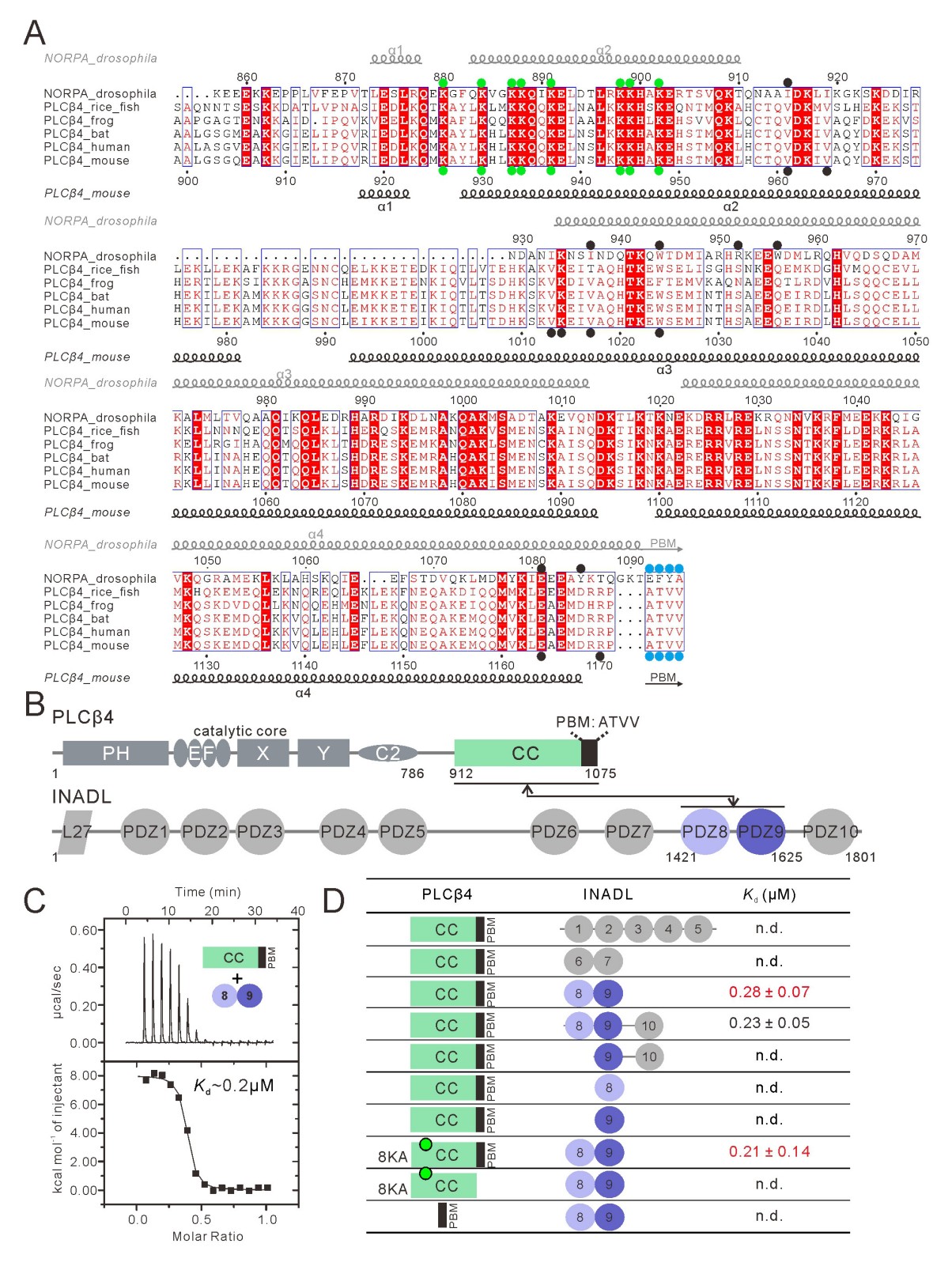

**Figure 4.** PLCβ4 and NORPA share a similar PDZ tandem binding mode. (**A**) Multiple sequence alignment of NORPA with PLCβ4 proteins from various animals by ClusterW and ESpript (espript.ibcp.fr/ESPript/ESPript/). Strictly conserved residues are boxed in white on a red background, and highly conserved residues are boxed in red on a white background. Helical structures as well as the PBM β-strand are depicted. The PBMs of NORPA and PLCβ4 are highlighted by blue dots. The key residues of the NORPA CC (or the PLCβ4 CC) involved in the interaction with INAD PDZ45 (or INADL

*Figure 4 continued on next page*

Figure 4 continued

PDZ89) are highlighted by black dots. The Lys residues mutated to Ala in order to facilitate crystallizations are highlighted by green dots. (B) Schematic diagram showing the domain organizations of PLCβ4 and INADL. The interaction is mediated by the PLCβ4 CC-PBM and INADL PDZ89 as indicated. The color coding of the domains is kept throughout this paper. (C) ITC-based measurement of the binding between the PLCβ4 CC-PBM and INADL PDZ89. (D) Table summarizing the measured dissociation constants between various constructs of PLCβ4 and INADL derived from ITC-based assays.
DOI: https://doi.org/10.7554/eLife.41848.011

The following figure supplements are available for figure 4:

**Figure supplement 1.** The PLCβ4 CC-PBM specifically interacts with INADL.
DOI: https://doi.org/10.7554/eLife.41848.012
**Figure supplement 2.** Characterization of the binding of INADL PDZ89 and the PLCβ4 CC-PBM.
DOI: https://doi.org/10.7554/eLife.41848.013

had no detectable binding to the PLCβ4 CC-PBM (*Figure 4D* and *Figure 4—figure supplement 2C&D*), indicating that the PDZ89 tandem is absolutely required for the formation of PLCβ4–INADL complex. Taken together, the biochemical results described above indicate that INADL PDZ89 specifically interacts with the PLCβ4 CC-PBM using a mode similar to that of the INAD PDZ45 and NORPA CC-PBM interaction.

## Structural characterization of the PLCβ4 CC-PBM–INADL PDZ89 complex

To elucidate the molecular mechanism governing the interaction between PLCβ4 and INADL, we determined the crystal structure of the INADL PDZ89–PLCβ4 CC-PBM complex. To facilitate the complex crystal growth, we also substituted the eight Lys residues in the α2 helix of PLCβ4 CC with Ala (*Figure 4A*). ITC analysis showed that the PLCβ4 CC$^{8KA}$-PBM binds to INADL PDZ89 with the same affinity as WT PLCβ4 CC-PBM (*Figure 4D* and *Figure 4—figure supplement 2E*). Furthermore, we discovered that covalently linking INADL PDZ89 with the PLCβ4 CC$^{8KA}$-PBM was necessary to obtain diffraction-quality crystals of the complex. The complex structure was determined by the single-wavelength anomalous dispersion method using gold derivatives at a resolution of 2.8 Å (*Supplementary file 1*). Strikingly, the overall architecture of the PLCβ4 CC$^{8KA}$-PBM–INADL PDZ89 complex is remarkably similar to that of the NORPA–INAD complex (*Figure 5A–C*). In the complex, PDZ89 forms an integral structural unit and extensively binds to the CC-PBM domain of PLCβ4, burying a total of ~1211 Å$^2$ surface area (*Figure 5A*). Structural analysis showed that the PDZ tandems and the corresponding contact region on PLCβ of the two complexes could be nicely aligned (*Figure 5C*, dashed box). In the two complexes, INADL PDZ9 can be nicely aligned with INAD PDZ5, and INADL PDZ8 needs to be rotated by ~43 degrees clockwise to superimpose with INAD PDZ4 (*Figure 5D–F*). The secondary structural elements that are involved in the inter-PDZ domain packing of the two PDZ tandems are quite similar (*Figure 5D* vs *Figure 5E*; *Figure 5—figure supplement 3*). The conformations of the C-terminal half of α3 and the N-terminal half of α4 of the two CC-PBMs are very different. The three helices (α2, α3, and α4) of the PLCβ4 CC form a flat sheet. By contrast, the α3 helix of the NORPA CC has an obvious kink in the center, resulting in a large curvature of the α3/α4 coiled coil (*Figure 5C*).

In the complex, the formation of a supramodule between PDZ8 and PDZ9 is mediated by a number of hydrophobic residues from both domains (*Figure 5H*). NMR-based analysis showed that, before binding to the PLCβ4 CC-PBM, there is minimal directly coupling between PDZ8 and PDZ9 (see *Figure 5—figure supplement 1A* for details). This result indicates that, unlike the tight coupling between PDZ4 and PDZ5 in INAD (*Liu et al., 2011*), the direct coupling between PDZ8 and PDZ9 is induced by the binding of PLCβ4. Mutations of the residues involved in PDZ89 inter-domain coupling led to decreased binding of the PDZ tandem to PLCβ4 CC-PBM (*Figure 5I* and *Figure 5—figure supplement 2B*). In INAD PDZ45, the PDZ5 C-terminal extension tucks into a groove at the rear of PDZ4, facilitating the formation of the PDZ45 supramodule (*Liu et al., 2011*). In INADL PDZ89, although the C-terminal extension of PDZ9 could not be resolved in the complex crystals, deletion of the C-terminal 10-residue extension of PDZ9 also weakened PDZ89's binding to the PLCβ4 CC-PBM (last row of *Figure 5I*, marked as 'ΔCT'), suggesting that the PDZ9 C-terminal extension may also participate in the formation of the PDZ89 supramodule when in complex with PLCβ4.

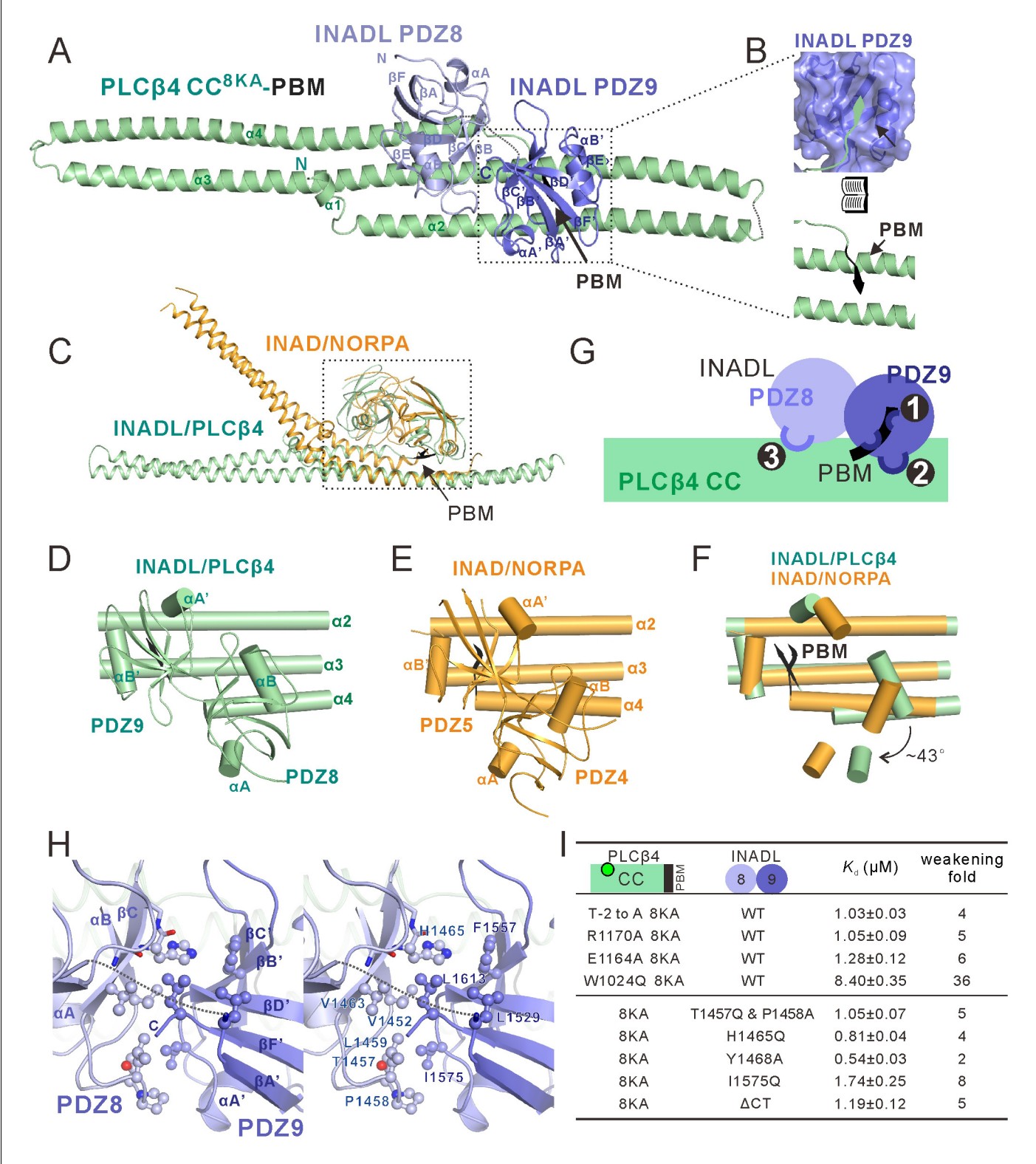

**Figure 5.** Structure of the PLCβ4 CC$^{8KA}$-PBM–INADL PDZ89 complex. (**A**) Ribbon representation of the PLCβ4 CC$^{8KA}$-PBM–INADL PDZ89 complex structure. The disordered loops are drawn as dashed lines in the ribbon representation. (**B**) Open-book view showing the interaction interface between INADL PDZ9 and the PBM of PLCβ4. (**C**) Superimposition of the NORPA CC$^{8KA}$-PBM–INAD PDZ45 complex structure (yellow) with the PLCβ4 CC$^{8KA}$-PBM–INADL PDZ89 complex structure (green), showing the similar binding mode of the two complexes. (**D–F**) Comparison of the PDZ tandem orientations in the PLCβ4 CC$^{8KA}$-PBM–INADL PDZ89 (**D**) and the NORPA CC$^{8KA}$-PBM–INAD PDZ45 (**E**) complexes. (**F**) The superposition of the CC$^{8KA}$-

*Figure 5 continued on next page*

Figure 5 continued

PBM portion of NORPA and PLCβ4 from the two complexes. Only the αA and αB helices of the PDZ domains are shown to illustrate the orientation differences of the PDZ domains in the two complexes. (G) Schematic cartoon diagram summarizing the interaction mode of the PLCβ4 CC⁸ᴷᴬ-PBM– INADL PDZ89 complex with three characteristic binding sites (Sites 1, 2, and 3; detailed in *Figure 5—figure supplement 1B and C*). (H) Stereoview showing the interaction in the interfaces between INADL PDZ8 and PDZ9. In this drawing, the side chains of the residues involved in the inter-domain interactions are drawn in the stick and ball representations. (I) A table summarizing the measured dissociation constants shows that mutations of the critical residues in the PDZ8 and PDZ9 interface can weaken the PLCβ4 CC⁸ᴷᴬ-PBM–INADL PDZ89 interaction.

DOI: https://doi.org/10.7554/eLife.41848.014

The following figure supplements are available for figure 5:

**Figure supplement 1.** Detailed analysis of the interaction between INADL PDZ89 and the PLCβ4 CC-PBM.

DOI: https://doi.org/10.7554/eLife.41848.015

**Figure supplement 2.** Impact of the mutations on the binding affinity between INADL PDZ89 and the PLCβ4 CC-PBM.

DOI: https://doi.org/10.7554/eLife.41848.016

**Figure supplement 3.** INADL PDZ89 instead of MUPP1 PDZ10-11 specifically interacts with PLCβ4.

DOI: https://doi.org/10.7554/eLife.41848.017

Like that of the INAD–NORPA complex, the binding interface of the INADL PDZ89–PLCβ4 CC-PBM complex can be divided into three distinct sites (*Figure 5G*): the PDZ9–PBM interaction site (site 1, *Figure 5G* and *Figure 5—figure supplement 1B*), the PDZ9/CC interaction site (site 2, *Figure 5G* and *Figure 5—figure supplement 1C*), and the PDZ8–CC binding site (site 3, *Figure 5G* and *Figure 5—figure supplement 1C*). In site 1, the last four amino acids (ATVV) of the PLCβ4 PBM bind to the canonical αB/βB-pocket of PDZ9 (*Figure 5B* and *Figure 5—figure supplement 1B*). In site 2, V-1 of the PLCβ4 PBM interacts with the evolutionarily conserved W1024 from α3 of the CC domain (*Figure 5—figure supplement 1C*). Multiple residues from α3 of the PLCβ4 CC contact with αB of PDZ9 through hydrophobic and charge–charge interactions (*Figure 5—figure supplement 1C*). R1170, situated immediately N-terminal to PLCβ4 PBM, forms hydrogen bonds with Y1468 and E1466 from βC of INADL PDZ8 (*Figure 5—figure supplement 1C*). In site 3, the carboxyl group of E1164 from α4 of the PLCβ4 CC forms hydrogen bonds with the backbone of the GLGL loop of PDZ8 by mimicking the classical carboxyl group–GLGF binding found in all PDZ–target interactions (*Figure 5—figure supplement 1C*). Mutations of the residues analyzed above invariably weakened the interaction between INADL PDZ89 and the PLCβ4 CC-PBM (top four rows in *Figure 5I*, and *Figure 5—figure supplement 2*).

Interestingly, MUPP1, a close homologue of INADL, shares a very high sequence homology with INADL (~82% sequence similarity, *Figure 5—figure supplement 3*) but has no detectable binding to PLCβ4, suggesting that the interaction between INADL and PLCβ4 is very specific. More strikingly, the residues from INADL PDZ89 that form the binding interface with PLCβ4 are identical in the corresponding positions in the MUPP1 PDZ10-11 tandem (highlighted in black dots in *Figure 5—figure supplement 3A*), but MUPP1 PDZ10-11 does not bind to the PLCβ4 CC-PBM (*Figure 5—figure supplement 3B1*). Detailed sequence analysis revealed that the set of residues that mediate the domain coupling of INADL PDZ89 (highlighted in blue dots in *Figure 5—figure supplement 3A*) are different in MUPP1 PDZ10-11. For example, the short linker that connects PDZ8 and PDZ9 of INADL is obviously different from the linker connecting PDZ10 and PDZ11 of MUPP1 (*Figure 5—figure supplement 3A*). Interestingly, substitution of the MUPP1 linker with the corresponding INADL linker (a 9-residue fragment highlighted in yellow box in *Figure 5—figure supplement 3A*) converted MUPP1 PDZ10-11 to a PLCβ4 CC-PBM binder, though still with a weak affinity (Kd ~80 μM) (*Figure 5—figure supplement 3B and C*). On the basis of this chimera, the conversion of Leu1635 in the βB/βC-loop of MUPP1 PDZ10 to Pro (corresponding to Pro1458 in the same position of INADL PDZ8; *Figure 5—figure supplement 3A*) further enhanced MUPP1 PDZ10-11's binding to the PLCβ4 CC-PBM (*Figure 5—figure supplement 3B and C*). The above analysis indicated that a small set of residues can determine the specific binding of INADL PDZ89 to the PLCβ4 CC-PBM.

## Discussion

In this study, we discover that the INAD PDZ45 supramodule binds to the CC-PBM domain of NORPA with very strong binding affinity. The strong binding between INAD and NORPA requires

the direct conformational coupling of PDZ4 and PDZ5 (i.e. formation of the PDZ45 supramodule) of INAD as well as precise spacing between the coiled-coil domain and the PBM of the NORPA. Such conformational arrangement provides an elegant way to synergistically integrate multiple interaction sites, each with relatively weak binding affinity, to form the very strong overall INAD–NORPA interaction. In addition, the stringent conformational requirement of both proteins for the tight binding also means that the interaction between INAD PDZ45 and the NORPA CC-PBM is highly specific. Finally, the sensitivity of the binding to the conformation of both INAD PDZ45 and the NORPA CC-PBM suggests that the interaction between INAD and NORPA may be regulated by altering the conformation of one or both proteins.

Earlier studies showed that, upon light activation, NORPA can rapidly hydrolyze $PIP_2$ and cause transient and localized acidification of microvilli, an event that can open the TRP channel (*Huang et al., 2010*) and disrupt the domain coupling between PDZ4 and PDZ5 of INAD PDZ45 (*Liu et al., 2011*). We demonstrated in this study that a mild acidification of the binding buffer from neutral pH to a mild acidic pH of 5.8 was sufficient to weaken the binding between INAD PDZ45 and NORPA CC-PBM dramatically (*Figure 3E*). Therefore, it is likely that the light-induced transient acidification of microvilli can trigger conformational uncoupling of the INAD PDZ45 supramodule, which acts as a negative feedback mechanism for rapid dissociation of NORPA from INAD and subsequent signal termination of each light-induced signaling cycle in *Drosophila* compound eyes. Therefore, INAD not only functions as a passive scaffold in assembling various components into a large signaling complex, but is also capable of actively modulating light signal transductions by regulating the dynamic organization of the entire light signaling complex.

Combination of the available genetic and biochemical data in the literature and the findings presented here leads us to propose an atomic model depicting the INAD-organized signaling complex beneath the *Drosophila* rhabdomeric membranes (*Figure 3D*). In the dark, the INAD PDZ345 tandem assembles a high-affinity, stoichiometric complex with the TRP channel and NORPA that is located right beneath the rhabdomeric membranes, thereby positioning NORPA in very close proximity to the TRP channel (*Figure 3D*). In addition, eye-PKC is also tethered to NORPA and TRP by binding INAD PDZ2. It is envisioned that, in this condition, the TRP channel is at its greatest sensitivity and efficiently senses the membrane lipid component changes initiated by light-induced $PIP_2$ hydrolysis by NORPA. In line with this model, disruption of the interaction between INAD and NORPA, either by mutation of INAD PDZ45 (in the *inaD$^2$* mutant flies) or by mutation of NORPA (Y-1 mutation), is known to cause severe functional defects in signal sensitivity and in kinetics in the photon-transduction pathway (*Cook et al., 2000*; *Shieh et al., 1997*; *Tsunoda et al., 1997*). Once the light activates the signaling cascade, the transient local acidification induced by the NORPA-mediated hydrolysis of $PIP_2$ can rapidly promote the structural uncoupling of PDZ45, thereby leading to the dissociation of the NORPA–INAD complex and termination of the light signal.

Different from mammal rod and cone opsin signaling, melanopsin photo-transduction is analogous to the *Drosophila* photo-transduction cascade. Activated by light, melanopsin can interact with a Gq/G11-type G protein (*Graham et al., 2008*), which in turn activates PLCβ4 (a vertebrate orthologue of NORPA in *Drosophila*) (*Jiang et al., 1994*; *Lee et al., 1994*). PLCβ4 hydrolyzes $PIP_2$ and thereby generates inositol 1,4,5 triphosphate ($Ins(1,4,5)P_3$) and diacylglycerol (DAG), which may ultimately modulate ion channels such as the TRPC6 or TRPC7 ion channels (*Xue et al., 2011*). Protein kinase C zeta (PKCζ), which is analogous to the eye-PKC in flies, may influence TRP channel activity via kinase-dependent phosphorylation(s) (*Peirson et al., 2007*). However, the molecular components and their action mechanisms in the ipRGCs photo signaling processes are far from certain. Given the sequence similarity between NORPA and PLCβ4, we speculate and also provide biochemical evidence to suggest that PLCβ4 may use a binding mode similar to that used by NORPA in ipRGC signaling. Biochemical characterizations revealed that PLCβ4 can specifically interact with PDZ89 of INADL. The complex structure of PLCβ4–INADL demonstrates that PLCβ4 interacts with the INADL PDZ89 supramodule with a mode very similar to that of the NORPA–INAD complex. Future studies will be required to test whether the INADL–PLCβ4 interaction identified in this study might be functional in ipRGC signaling or in PLCβ4 signaling in other tissues.

Finally, as a technical note, we struggled for a long time to obtain optimal crystals of the INAD–NORPA complex for structural analysis. It was a bitter but ultimately sweet lesson to learn that systematic substitution of eight Lys residues on one helix of the coiled coil of NORPA by Ala produced diffraction-quality crystals due to introduction of a new crystal packing surface. The elongated

conformation of coiled coils provides fewer favorable surface areas for crystal packing, and thus coiled coils are often more difficult to crystalize than to globular proteins. To assist coiled-coil protein crystallization, protein engineering tricks such as surface entropy reduction, site-directed mutations (*Wine et al., 2009*), construct optimization to deplete the disordered sequences (*Zhou et al., 2017*), introducing new packing surface by replacing charged amino acid acids with small hydrophobic and helix favoring Ala (this study) may be useful strategies.

# Materials and methods

### Key resources table

| Reagent type (species) or resource | Designation | Source or reference | Identifiers | Additional information |
|---|---|---|---|---|
| Strain, strain background (E. coli) | BL21(DE3) | Novagen | Cat #69450 | |
| Strain, strain background (*Escherichia coli*) | B834(DE3) | Novagen | Cat #69041 | |
| Strain, strain background (E. coli) | Rosseta (DE3) | Novagen | Cat #70954 | |
| Cell line (human) | HEK293T | ATCC | Cat #CRL-3216; RRID:CVCL_0063 | |
| Transfected construct (plasmid) | GFP-INAD *Drosophila* PDZ45 WT | This paper | N/A | |
| *Transfected construct (plasmid)* | *GFP-INAD Drosophila PDZ45 G605E* | This paper | N/A | |
| *Transfected construct (plasmid)* | *GFP-INAD Drosophila PDZ45 T669E* | This paper | N/A | |
| Recombinant protein | *Drosophila INAD FL WT: aa M1-A674* | This paper | NCBI: NM_166566.1 | |
| Recombinant protein | *Drosophila* INAD PDZ45 WT: aa K473-A674 | This paper | NCBI: NM_166566.1 | |
| Recombinant protein | *Drosophila* INAD PDZ5 WT: aa L580-P665 | This paper | NCBI: NM_166566.1 | |
| Recombinant protein | *Drosophila* NORPA CC-PBM WT: aa E863-A1095 | This paper | NCBI: NM_080330.4 | |
| Recombinant protein | *Drosophila* NORPA CC WT: aa E863-T1092 | This paper | NCBI: NM_080330.4 | |
| Recombinant protein | *Drosophila* NORPA CC[8KA]-PBM: aa E863-A1095; K880A and K884A and K887A and K888A and K891A and K898A and K899A K902A | This paper | NCBI: NM_080330.4 | |

*Continued on next page*

*Continued*

| Reagent type (species) or resource | Designation | Source or reference | Identifiers | Additional information |
|---|---|---|---|---|
| Recombinant protein | *Drosophila* NORPA CC-PBM K-5(GSGS)T-4: aa D852-K1090, GSGS, T1091-A1095 | This paper | NCBI: NM_080330.4 | |
| Recombinant protein | Human INADL PDZ1-5 WT: aa M126-D776 | This paper | NCBI: NM_176877.3 | |
| Recombinant protein | Human INADL PDZ6-7 WT: aa G1054-P1332 | This paper | NCBI: NM_176877.3 | |
| Recombinant protein | Human INADL PDZ89 WT: aa L1421-T1625 | This paper | NCBI: NM_176877.3 | |
| Recombinant protein | Human INADL PDZ8-10 WT: aa S1422-D1801 | This paper | NCBI: NM_176877.3 | |
| Recombinant protein | Human INADL PDZ9-10 WT: aa E1530-D1801 | This paper | NCBI: NM_176877.3 | |
| Recombinant protein | Human INADL PDZ8 WT: aa S1422-N1528; | This paper | NCBI: NM_176877.3 | |
| Recombinant protein | Human INADL PDZ9 WT: aa E1530-R1659 | This paper | NCBI: NM_176877.3 | |
| Recombinant protein | Human INADL PDZ89 ΔCT: aa L1421-R1615 | This paper | NCBI: NM_176877.3 | |
| Recombinant protein | Mouse PLCβ4 CC-PBM WT: aa E912-V1175 | This paper | NCBI: NM_013829.2 | |
| Recombinant protein | Mouse PLCβ4 CC[8KA] –PBM: aa: E912-V1175; K926A and K930A and K933A and K934A and K937A and K944A and K945A and K948A | This paper | NCBI: NM_013829.2 | |
| *Recombinant protein* | Mouse PLCβ4 CC[8KA]: aa E912-A1172[8KA] | This paper | NCBI: NM_013829.2 | |
| *Recombinant protein* | INADL-PLCβ4 fusion: human INADL aa L1421-T1625, GGGGSGGG GSGGEGS, mouse PLCβ4[8KA] aa E912-V1175[8KA] | This paper | NCBI: NM_176877.3; NM_013829.2 | |
| *Recombinant protein* | Mouse GRIP1 PDZ1-3 WT: aa M1-A334 | This paper | NCBI: NM_130891.2 | |
| *Recombinant protein* | Rat GRIP1 PDZ4-7 WT: aa Q463-N1069 | This paper | NCBI: NM_032069.1 | |
| *Recombinant protein* | Human NHERF1 FL WT: aa M1-L358 | This paper | NCBI: NM_004252.4 | |

*Continued on next page*

*Continued*

| Reagent type (species) or resource | Designation | Source or reference | Identifiers | Additional information |
|---|---|---|---|---|
| *Recombinant protein* | Human NHERF2 FL WT: aa M1-F337 | This paper | NCBI: NM_001130012.2 | |
| *Recombinant protein* | Mouse MUPP1 PDZ10-11 WT: aa G1610-P1802 | This paper | NCBI: NM_001305284.1 | |
| Synthesized peptide | Mouse PLCβ4 PBM 1166EMDRR PATVV1175 | Synthesized by Chinapeptides Co. Ltd. | N/A | |
| Synthesized peptide | *Drosophila* NORPA PBM 1086KTQGK TEFYA1095 | Synthesized by Chinapeptides Co. Ltd. | N/A | |
| Antibody | Anti-GFP (B-2) (mouse mAb) | S anta Cruz Biotechnology | Cat# sc-9996; RRID: AB_627695 | Dilution factor: 1:1000 |
| Antibody | Anti-Mouse IgG (Goat polyAb) | Sigma | Cat# A4416; RRID: AB_258167 | Dilution factor: 1:20000 |
| Commercial assay or kit | Clone Express II, One-Step Cloning Kit | Vazyme Biotech Co., Ltd | Cat #C112 | |
| Commercial assay or kit | ViaFect transfection reagent | Promega Corporation | Cat #E4981 | |
| Software, algorithm | Origin7.0 | OriginLab | http://www.originlab.com/; RRID: SCR_002815 | ITC titration data analysis |
| Software, algorithm | GraphPad Prism | GraphPad Software Inc. | http://www.graphpad.com/scientific-software/prism; RRID: SCR_002798 | FITC titration data analysis |
| Software, algorithm | HKL2000 | HKL Research Inc. | http://www.hkl-xray.com/ | Diffraction data processing and scaling |
| Software, algorithm | CCP4 | PMID: 21460441 | http://www.ccp4.ac.uk/; RRID: SCR_007255 | Crystal structure determination |
| Software, algorithm | PHENIX | PMID: 20124702 | http://www.phenix-online.org/; RRID: SCR_014224 | Model building and refinement |
| Software, algorithm | PyMOL | DeLano Scientific LLC | http://www.pymol.org/; RRID: SCR_000305 | Structure figure plot |
| Software, algorithm | ASTRA6.1 | Wyatt Technology Corporation | http://www.wyatt.com/products/software/astra.html | Light-scattering data analysis |
| Software, algorithm | NMRPipe | NIH | https://spin.niddk.nih.gov/NMRPipe/ref/index.html | NMR data processing |
| Software, algorithm | Sparky | UCSF Sparky | https://www.cgl.ucsf.edu/home/sparky/ | NMR data analysis |

## Cloning and constructs for recombinant protein expression

cDNA encoding PDZ5 (residues 580–665) and PDZ45 (residues 473–674) were PCR amplified from *Drosophila melanogaster inaD* and cloned into a modified version of the pET32a vector.

The NORPA CC-PBM was PCR amplified from the cDNA of *Drosophila* NORPA (from *Drosophila* Genomics Resource Center, NCBI reference sequence: NP_525069.2, 1095aa) and cloned into a modified version of the pET32a vector containing an N-terminal 6 × His tag.

PLCβ4 cDNA encoding residues 912–1175 was PCR amplified from mouse *PLCβ4* plasmid (Genebank: NM_013829.2). INADL cDNA encoding PDZ89 (residues 1421–1625) was PCR amplified from human *INADL* (Genebank: NM_176877.3). Various mutations of PLCβ4 or INADL89 were generated using the standard PCR-based method and confirmed by DNA-sequencing. MUPP1 PDZ10-11 cDNA encoding residues 1610–1802 was PCR amplified from mouse *MUPP1* (NM_001305284.1). INADL PDZ89 was fused to the N-terminus of PLCβ4 CC-PBM with a 15aa linker (GGGGSGGGGSGGEGS) to generate a fusion protein used for crystallization.

## Protein expression and purification

Recombinant proteins were expressed in *Escherichia coli* BL21 (DE3) or Rosseta (DE3) host cells at 16°C. The His-tagged fusion proteins were purified by $Ni^{2+}$-NTA-Sepharose (GE) affinity chromatography followed by size-exclusion chromatography (Superdex 200 column from GE Healthcare) in the final buffer of 50 mM Tris·HCl, 1 mM DTT, 1 mM EDTA, pH 7.5, and 100 mM NaCl. When needed, the N-terminal His-tagged was cleaved by protease 3C and removed by another step of size-exclusion chromatography. To obtain highly purified PLCβ4 CC-PBM, a cation-exchange chromatography was used after the size-exclusion chromatography. Uniformly $^{15}$N-labeled INAD PDZ45 and INADL PDZ89 were prepared by growing bacteria in M9 minimal medium using $^{15}$NH$_4$Cl (Cambridge Isotope Laboratories Inc.) as the sole nitrogen source. Se-Met-labeled NORPA CC-PBM was prepared by expressing the protein in B834 (DE3) host cells grown in Se-Met supplemented M9 minimal medium at 37°C for ~13 hr and purified by size-exclusion chromatography coupled with cation-exchange chromatography as described above.

Purified isolated NORPA CC-PBM protein (1 mg/ml in a buffer of 50 mM HEPES pH 7.6, 100 mM NaCl, 1 mM DTT and 1 mM EDTA) was reductive methylated for crystallization. Stock solutions of 1 M dimethylamine-borane complex (ABC) and 1 M formaldehyde in water were freshly prepared and kept at 4°C or on ice (*Walter et al., 2006*). To start the methylation reaction, 20 μL ABC and 40 μl formaldehyde were added to each ml of NORPA CC-PBM solution, and the reactions were continued for 2 hr at 4°C before another 20 μL ABC and 40 μl formaldehyde was added to the reaction mixture. The reaction was continued for another 2 hr, and finally an additional 10 μl of ABC was added and the reaction was allowed to continue overnight at 4°C. To quench the reaction, the reaction mixture was passed through a size-exclusion column equilibrated with a 50 mM Tris buffer (pH 7.5) containing 100 mM NaCl, 1 mM DTT and 1 mM EDTA.

## Crystallography

Crystals of the Se-Met NORPA CC-PBM, with K884E and K898E substitutions, were obtained by the hanging drop vaporing diffusion method at 16°C. In each well, 1 μL purified Se-Met NORPA CC-PBM$^{K884E \& K898E}$ at 27 mg/mL was mixed with 1 μL crystallization buffer. Crystals formed within two days from the solution containing 64% 2-methyl-2,4-pentanediol (MPD), 100 mM HEPES (pH 7.2), and 8% pentaerythritol ethoxylate (3/4 EO/OH). Crystals of methylated NORPA CC-PBM were grown in buffer containing 3.8 M sodium formate and 100 mM bis-tris propane at pH 6.6. The crystals of the NORPA CC$^{8KA}$-PBM–INAD PDZ45 complex (~10 mg/mL in 50 mM Tris (pH 7.8), 100 mM NaCl, 1 mM EDTA, and 1 mM DTT buffer) were grown in reservoir solution containing 0.2 M MgCl$_2$, 0.1 M Tris pH 6.5, and 25% w/v polyethylene glycol 3350. Crystals of the PLCβ4 CC$^{8KA}$-PBM–INADL PDZ89 complex (~10 mg/mL in 50 mM Tris (pH 7.8), 100 mM sodium chloride, 1 mM EDTA, and 1 mM DTT buffer) were grown in reservoir solution containing 0.2 M MgCl$_2$, 0.1 M Bis-Tris (pH 6.5), and 20% w/v polyethylene glycol 3350. To prepare Au-derivatives, crystals were soaked in crystallization solution containing 25 mM KAu(CN)$_2$ for 1 to 2 d. Crystals were then soaked in reservoir solution containing an extra 10% (vol/vol) glycerol for cryo-protection.

Diffraction data were collected at the Shanghai Synchrotron Radiation Facility (BL17U or BL19U1) at 100 K. Data were initially processed and scaled using HKL2000 or HKL3000 (*Otwinowski and Minor, 1997*). The data from the PLCβ4 CC$^{8KA}$-PBM–INADL PDZ89 complex were further corrected for anisotropy using the diffraction anisotropy server (https://services.mbi.ucla.edu/anisoscale/) and truncated to 3.0 Å, 3.1 Å and 2.8 Å along the a, b and c axes, respectively. In the structure

determinations processes, different strategies were used for different crystals. The Se-Met NORPA CC-PBM$^{K884E\ \&\ K898E}$ and PLCβ4 CC$^{8KA}$-PBM–INADL PDZ89 complex structures were determined using the single-wavelength anomalous dispersion (SAD) method. The Se or gold sites were found by SHELXD (*Dall'Antonia et al., 2003*). Subsequent site refinement, phase calculation, density modification and initial model building were carried out with Autosol (*Terwilliger et al., 2009*). Molecular replacement was carried out for the methylated NORPA CC-PBM and the NORPA CC$^{8KA}$-PBM–INAD PDZ45 complex, using the N-terminal halves of the NORPA CC-PBM$^{K884E\ \&\ K898E}$ structure and the INAD PDZ45 structure (PDB ID: 3R0H) as the search models for PHASER (*McCoy et al., 2007*). Subsequent model building and refinement for all of the four different structures were completed iteratively using COOT (*Emsley et al., 2010*) and PHENIX (*Adams et al., 2010*). The final models were validated by MolProbity (*Chen et al., 2010*). The final refinement statistics are summarized in *Supplementary file 1*. All structure figures were prepared by PyMOL (www.pymol.org). The sequence alignments were prepared and presented using ClustalW (*Larkin et al., 2007*) and ESPript (*Robert and Gouet, 2014*), respectively. The structure factors and the coordinates of the structures reported in this work have been deposited to PDB under the accession codes of 6IRB for the Se-Met NORPA CC-PBM with K884E and K898E substitutions; 6IRC for the methylated NORPA CC-PBM; 6IRE for the NORPA CC$^{8KA}$-PBM–INAD PDZ45 complex; and 6IRD for the PLCβ4 CC$^{8KA}$-PBM–INADL PDZ89 complex.

## Isothermal titration calorimetry assay

ITC measurements for INAD–NORPA interactions were carried out on VP-ITC or ITC200 Microcal calorimeters (Microcal) at 25°C. For INADL–PLCβ4 interactions, ITC measurements were carried out on an ITC200 Microcal calorimeter (Microcal) at 16°C. The titration buffer contained 50 mM Tris-HCl (pH 7.5), 1 mM DTT, 1 mM EDTA, and 100 mM NaCl. Each titration point was performed by injecting a 10 μL (for VP-ITC) or 2.5 μL (for ITC200) aliquot of a protein sample from a syringe into a protein sample in the cell at a time interval of 120 s to ensure that the titration peak returned to the baseline. The titration data were analyzed by Origin7.0 (Microcal).

## NMR spectroscopy

NMR samples contained 0.2 mM of INAD PDZ45 or INADL PDZ89 in 50 mM Tris-HCl (pH 7.2, with 1 mM DTT and 1 mM EDTA) in 10% D$_2$O. NMR spectra were acquired at 30°C on Varian Inova 750- or 800-MHz spectrometers, each equipped with an actively z-gradient shielded triple resonance probe. The backbone resonance assignment of INAD PDZ45 was obtained using the data from our previous study (*Liu et al., 2011*).

## Pull-down assay

HEK293T cells were cultured in Dulbecco's Modified Eagle Medium (DMEM) supplemented with 10% fetal bovine serum (FBS), and 1% of penicillin-streptomycin at 37°C with 5% CO$_2$. Cells were tested negative for mycoplasma contamination by cytoplasmic DAPI staining. The cell line was only used for heterologous protein expression, so no further authentication was performed. After reaching 70–80% confluency, cells were transfected with ViaFect transfection reagent (Promega) using GFP control, GFP-INAD PDZ45 (aa 473–674), or the G605E and T669E mutants of INAD PDZ45. The cells were collected 48 hr after transfection. HEK293T cell lysate expressing GFP control, GFP-INAD PDZ45, or the G605E and T669E mutants of GFP-INAD PDZ45 was incubated with 0.5 nmol of purified His-StrepII-tagged NORPA CC-PBM (aa 863–1095) for 40 mins at 4°C. Each mixture was incubated with 20 μl StrepTactin Sepharose High Performance slurry beads (GE Healthcare) for another 1 hr at 4°C before pelleting by centrifugation. After washing twice with buffer (50 mM Tris (pH 7.5), 100 mM NaCl, 1 mM EDTA and 5 mM DTT), the captured proteins in the pellets were eluted by boiling with 2 × SDS loading buffer, resolved by 12% SDS-PAGE, and detected by immunoblotting with anti-GFP antibody (SANTA CRUZ, GFP-B2: sc-9996).

## Fluorescence assay

Fluorescence assays were performed on a PerkinElmer LS-55 fluorimeter at 25°C. In the assay, FITC (Molecular Probes)-labeled peptide samples were titrated with binding partners in 50 mM Tris buffer

(with pH value and DTT concentration specifically indicated) containing 50 mM NaCl and 1 mM EDTA. The titration curves were fitted with the GraphPad Prism five software package.

## Analytical gel-filtration chromatography

Protein samples (typically 100 µL at a concentration of 20 µM pre-equilibrated with the column buffer) were injected into an AKTA FPLC system with a Superose 12 10/300 GL column (GE Healthcare) using the column buffer of 50 mM Tris-HCl (pH 7.8), 1 mM DTT, 1 mM EDTA, and 100 mM NaCl.

## Liposome-binding assay

The total bovine brain lipid extracts (Folch fraction I, Sigma B1502) were re-suspended by sonication in a buffer containing 50 mM Tris (pH 7.5), 100 mM NaCl, 1 mM EDTA, and 1 mM DTT) at 10 mg/ml. The sedimentation-based assay followed the method described earlier (*Wen et al., 2008*). The liposome solution was pre-cleared by centrifugation at 20,817 *g*, 4°C for 2 min and the protein samples were also pre-cleared at 200,000 *g*, 4°C for 30 min. The protein samples (5 µM) were incubated with different concentrations of liposomes in 40 µl of buffer for 20 mins at room temperature and then centrifuged at 100,000 *g*, 4°C for 30 min in a Beckman TLA100.1 rotor. The supernatants were transferred into a new tube to determine the amount of proteins that did not bind to the liposomes. The pellets were washed twice with buffer and re-suspended in a volume of buffer equal to that of the supernatants. Finally, the proteins recovered from the supernatants and pellets were analyzed by SDS-PAGE.

## Acknowledgements

We thank the Shanghai Synchrotron Radiation Facility (SSRF) BL17U1 and BL19U1 for X-ray beam time. This work was supported by grants from the Minister of Science and Technology of China (2014CB910204), the National Key R and D Program of China (2016YFA0501903), the Natural Science Foundation of Guangdong Province (2016A030312016), a Shenzhen Basic Research Grant (JCYJ20160229153100269), RGC of Hong Kong (AoE-M09-12 and C6004-17G), and the Asia Fund for Cancer Research to MZ; grants from the National Natural Science Foundation of China (No. 31670765 and 31870746) and Shenzhen Basic Research Grants (JCYJ20160427185712266 and JCYJ20170411090807530) to WL; a GRF grant from RGC of Hong Kong (16104518) to FY; and grants from the Strategic Priority Research Program of the Chinese Academy of Sciences (XDA16020603, XDPB10, and XDB02010000), the National Natural Science Foundation of China (81790644), and the National Key Basic Research Program of China (2016YFA0400900) to TX. MZ is a Kerry Holdings Professor of Science and a Senior Fellow of the Institute for Advanced Study (IAS) at the Hong Kong University of Science and Technology (HKUST).

## Additional information

### Competing interests

Mingjie Zhang: Reviewing editor, *eLife*. The other authors declare that no competing interests exist.

### Funding

| Funder | Grant reference number | Author |
| --- | --- | --- |
| Research Grants Council, University Grants Committee | 16104518 | Fei Ye |
| National Natural Science Foundation of China | 81790644 | Tian Xue |
| National Natural Science Foundation of China | 31870746 | Wei Liu<br>Yuqian Ma |
| Chinese Academy of Sciences Key Project | XDA16020603 | Tian Xue |

| National Key Basic Research Program of China | 2016YFA0400900 | Tian Xue |
| Chinese Academy of Sciences Key Project | XDPB10 | Tian Xue |
| Chinese Academy of Sciences Key Project | XDB02010000 | Tian Xue |
| National Natural Science Foundation of China | 31670765 | Wei Liu |
| Shenzhen Basic Research Grant | JCYJ20170411090807530 | Wei Liu |
| Shenzhen Basic Research Grant | JCYJ20160427185712266 | Wei Liu |
| Shenzhen Basic Research Grant | JCYJ20160229153100269 | Mingjie Zhang Wei Liu |
| Ministry of Science and Technology | 2014CB910204 | Mingjie Zhang |
| Research Grants Council, University Grants Committee | AoE-M09-12 | Mingjie Zhang |
| Natural Science Foundation of Guangdong Province | 2016A030312016 | Mingjie Zhang |
| Research Grants Council, University Grants Committee | C6004-17 | Mingjie Zhang |
| National Key R&D Program of China | 2016YFA0501903 | Mingjie Zhang |

The funders had no role in study design, data collection and interpretation, or the decision to submit the work for publication.

### Author contributions

Fei Ye, Formal analysis, Investigation, Methodology, Writing—original draft, Writing—review and editing; Yuxin Huang, Jianchao Li, Formal analysis, Investigation; Yuqian Ma, Chensu Xie, Zexu Liu, Xiaoying Deng, Investigation; Jun Wan, Formal analysis, Funding acquisition; Tian Xue, Formal analysis, Supervision, Funding acquisition; Wei Liu, Formal analysis, Supervision, Investigation, Methodology, Writing—original draft, Writing—review and editing; Mingjie Zhang, Conceptualization, Supervision, Funding acquisition, Investigation, Writing—original draft, Project administration, Writing—review and editing

### Author ORCIDs

Fei Ye (iD) https://orcid.org/0000-0002-4268-7256
Jianchao Li (iD) http://orcid.org/0000-0002-8921-1626
Wei Liu (iD) http://orcid.org/0000-0001-8250-2562
Mingjie Zhang (iD) http://orcid.org/0000-0001-9404-0190

### Decision letter and Author response

Decision letter https://doi.org/10.7554/eLife.41848.035
Author response https://doi.org/10.7554/eLife.41848.036

## Additional files

### Supplementary files

• Supplementary file 1: Statistics of data collection and model refinement for NORPA CC-PBM structures, INAD–NORPA complex structure and INADL–PLCβ4 complex structure.
DOI: https://doi.org/10.7554/eLife.41848.018

• Transparent reporting form
DOI: https://doi.org/10.7554/eLife.41848.019

## Data availability

Diffraction data have been deposited at PDB under the accession numbers 6IRB, 6IRC, 6IRD, and 6IRE.

The following datasets were generated:

| Author(s) | Year | Dataset title | Dataset URL | Database and Identifier |
|---|---|---|---|---|
| Ye F, Li J, Yuxin Huang, Wei Liu, Mingjie Zhang | 2018 | Complex structure of INADL PDZ89 and PLCb4 C-terminal CC-PBM | http://www.rcsb.org/structure/6IRD | Protein Data Bank, 6IRD |
| Ye F, Li J, Liu Z, Chensu Xie, Wei Liu, Mingjie Zhang | 2018 | C-terminal domain of Drosophila phospholipase beta NORPA, methylated | http://www.rcsb.org/structure/6IRC | Protein Data Bank, 6IRC |
| Ye F, Li J, Liu Z, Chensu Xie, Wei Liu, Mingjie Zhang | 2018 | C-terminal coiled coil domain of Drosophila phospholipase C beta NORPA, selenomethionine | http://www.rcsb.org/structure/6IRB | Protein Data Bank, 6IRB |
| Ye F, Li J, Liu Z, Xiaoying Deng, Wei Liu, Mingjie Zhang | 2018 | Complex structure of INAD PDZ45 and NORPA CC-PBM | http://www.rcsb.org/structure/6IRE | Protein Data Bank, 6IRE |

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
