## [Decision Letter]

Thank you for submitting your article "An unexpected PDZ tandem-mediated PLCβ binding critical for photo signal transductions from *Drosophila* to mammals" for consideration by *eLife*. Your article has been reviewed by Richard Aldrich as the Senior Editor, a Reviewing Editor, and three reviewers. The reviewers have opted to remain anonymous.

The reviewers have discussed the reviews with one another and the Reviewing Editor has drafted this decision to help you prepare a revised submission.

Summary:

In this study, the authors discovered that the INAD PDZ4-PDZ5 module binds to the CC-PBM domain of NORPA with high affinity. The strong binding between INAD and NORPA requires interaction between PDZ4 and PDZ5 of INAD. Considering primary sequence similarity between NORPA and PLCβ4, the authors speculated and then provided biochemical, structural and functional evidence that PLCβ4 uses a similar binding mode as NORPA does in ipRGC signaling. Remarkably, the complex structure of PLCβ4/INADL revealed that PLCβ4 interacts with INADL PDZ89 module with a mode very similar to that of the NORPA/INAD complex.

Essential revisions:

1) As the authors note, the affinity between INAD PDZ45 and NORPA CC-PBM is among the tightest in all known PDZ/target interactions. Does the structure suggest possible reasons for the high affinity of this interaction?

2) The interaction interface between the INAD PDZ4 and PDX5 domains in the INAD/NORPA complex has been described in a previous publication (Liu et al.). Does this interface change at all when INAD is bound to NORPA? It would be interesting (but optional) to test the effect of mutations of this interface on binding to NORPA.

3) Does the interface between PDZ4 and PDZ5 show any, even remote, similarities that between PDZ8 and PDZ9 in the INADL/PLCβ4 interface?

4) The authors found that decreasing the pH of the buffer from 7.8 to 5.8 dramatically weakened the binding between INAD and NORPA by ~ 6700 fold (Figure 3E). Can this be rationalized by the structure of the complex?

5) The supramodule interaction between CC-PBM and PDZ tandem is supported by solid data and represents a novel functional mechanism of PDZ scaffold proteins. A potential concern is the involvement of this interaction in ipRGC signaling. When the PBM of NORPA is deleted, both activation and deactivation of the fly photo response becomes slow at all light intensities (Shieh et al., 1997). In the case of PLCβ4, in contrast, deletion of PBM did not cause any change in the photo response of ipRGC at high intensity, but slightly increased the deactivation speed of responses to dim light. This effect, if real, indicated a role opposite to the known purpose of assembling GPCR signaling complexes, i.e., to speed up the signaling. Is INADL actually required for the photo response of ipRGC? Have the authors tried to downregulate the expression of INADL, or at least show it colocalizes with PLCβ4, in ipRGC?

6) INADL may form signaling complex with PLCβ4 in some cell types, but the occurrence in ipRGC is not convincing at all based on the shown electrophysiological results. Please consider taking out this part of conclusion. If the authors would like to make the manuscript stronger, they could obtain in vivo evidence for the supramoldule interaction using fly assays. For instance, the authors could find a point mutation in PDZ4 of INAD that disrupts its NORPA interaction and demonstrate in transgenic flies that this mutation phenocopies PBM-lacking NORPA.

7) The new structures provide insight of interaction of PDZ domains and their binding proteins. However, the in vivo function of this interaction is lacking, and the conclusion that interactions of PLCβs/PDZ is conserved in ipRGC signaling is not fully supported. It has been previously shown that PDZ1 binds to the C-terminus of NORPA, while PDZ5 binds to an internal region (including in C-terminal coiled-coil domain) (van Huizen et al., 1998), moreover, the crystal structure of the PDZ1 and C-terminal peptide has been determined (Kimple et al., 2001). The authors found that PDZ45 bind strongly to C-terminal regions. Please comment and discuss.

8) Several key components of fly phototransduction are assembled by 5-PDZ domain protein, INAD, which supporting speed and sensitivity of fly vision. However, although the ipRGC signaling shares high similarity with phototransduction, this signal does not appear to demand the high sensitivity and signal rate. Therefore, it is a question if an INAD-like signal complex exists in the ipRGC signaling. There is no direct evidence that INADL has an in vivo function in the ipRGC signaling as INAD in fly vision. In fly photoreceptor cells, the C-terminal mutations of NORPA caused reduction of NORPA protein in rhabdomeres and severe defects in phototransduction as by slow activation and deactivation. Please comment and discuss.

9) Comments on the structures:

a) The R_free_ values seem too high for all four structures. Please check the refinements.

b) Please show a difference map of the PDZ45 module after molecular replacement phasing of the NORPA CC^8KA^-PBM/INAD PDZ45 (i.e., before the PDZ45 module was modeled).

c) The authors note that covalently linking INADL PDZ89 with PLCβ4 CC^8KA^-PBM was necessary to obtain diffraction quality crystals of the complex. Please elaborate. Was there no diffraction in the absence of the linker? Also, please provide details of the cross-linking procedure.

[Editors' note: further revisions were requested prior to acceptance, as described below.]

Thank you for resubmitting your work entitled "An unexpected PDZ tandem-mediated PLCβ binding critical for photo signal transductions in *Drosophila*” for further consideration at *eLife*. Your revised article has been favorably evaluated by Richard Aldrich (Senior Editor), a Reviewing Editor (Axel Brunger), and three reviewers.

The manuscript has been improved but there are some remaining issues that need to be addressed before acceptance, as outlined below:

The omit map (Author response image 6 in the rebuttal) with the entire PDZ45 module omitted is rather fragmented since it represents half of the entire structure. Please generate a simulated annealing composite omit map instead. Also, this map and representative portions of the final 2mFo-DFc map should be included in a supplementary figure.

In the previous version, the authors claimed that the new PDZ-PLC interaction mediated melanopsin signaling in mammalian ipRGC cells despite of some conflicting data. Although this conclusion and the related data have been removed in this revised version, the Introduction and Result sections still emphasize the potential role of this interaction in ipRGC signaling (for example, the Title, second paragraph of the Introduction and subsection “PLCβ4 adopts a similar mode in binding to INADL as NORPA does to INAD”). Please remove these parts, and rather discuss a potential involvement of a PDZL/PLC4 complex in ipRGC signaling at the end of Discussion.

The Title should also be softened in order avoid an unproven claim, i.e., that the binding is critical for *Drosophila* photo signaling.

---

## [Author Response]

Essential revisions:1) As the authors note, the affinity between INAD PDZ45 and NORPA CC-PBM is among the tightest in all known PDZ/target interactions. Does the structure suggest possible reasons for the high affinity of this interaction?

The complex structure of INAD PDZ45/NORPA CC-PBM indeed provides a molecular basis for the high affinity interaction between the two proteins as shown in Figure 2. In addition to the canonical PDZ/target interaction (i.e. with only several extreme C-terminal tail PDZ binding motif engaging PDZ domain; Site 1 in Figure 2D,E), INAD PDZ45/NORPA CC-PBM complex contains two additional binding sites (Sites 2 and 3 in Figure 2D,F,G). These two additional sites dramatically enhanced the binding (from *K*_d_ of 39 mM with Site 1 to 0.016 mM with three sites together; Figure 1D). The structure of the INAD PDZ45/NORPA CC-PBM complex also showed that the high order structure of the PDZ45 tandem and the NORPA coiled-coil are required for the binding (Figure 2), indicating that the interactions provided by Sites 2 and 3 not only enhance the affinity but also raise specificity of the INAD/NORPA binding.

2) The interaction interface between the INAD PDZ4 and PDX5 domains in the INAD/NORPA complex has been described in a previous publication (Liu et al.). Does this interface change at all when INAD is bound to NORPA? It would be interesting (but optional) to test the effect of mutations of this interface on binding to NORPA.

In the structure of INAD PDZ45 in complex with NG2 peptide (Liu et al., 2011), the contact between PDZ4 and PDZ5 is very extensive, includes the βB, βC, and the αA/βD loop, and βE/αB loop from PDZ4; also βB’, βB’/βC’ loop, βC’ and βD’ from PDZ5. The C-terminal tail of PDZ5 folds back and binds into a groove at the surface of PDZ4, and “staples” the two PDZ domains together (presented as Figure 5D in Liu et al. (2011)). We have compared the structures of INAD PDZ45 in complex with NG2 peptide and in complex with NORPA CC-PBM in the Figure 2C of our manuscript. The packing interface of INAD PDZ4 and PDZ5 in the two complexes are essentially the same (this point has been added in the revised manuscript). Based on the complex structure, we believe that PDZ45 coupling is important for the high-affinity binding to NORPA CC-PBM. To test this hypothesis, we generated a PDZ45 mutant with Thr669, a residue required for the inter-domain stability, substituted by Glu (T669E-PDZ45). The T669E mutant was previously shown to disrupt PDZ45 supramodule formation (Liu et al., 2011). Pulldown analysis revealed the T669E-PDZ45 mutant has a much weaker binding to NORPA CC-PBM (Author response image 1), indicating that the formation of PDZ45 supramodule is important for binding to NORPA. We have included this data as Figure 2—figure supplement 5D in the revised manuscript.

**Author response image 1. respfig1:** Pull-down assay showing that the T669E mutant of INAD PDZ45 impairs its binding to NORPA CC-PBM. In this assay, GFP-fused T669E-INAD PDZ45 or WT INAD PDZ45 was expressed in HEK293T cells, and pulled down by purified His-StrepII-tagged NORPA CC-PBM as detailed in the Materials and methods section.

3) Does the interface between PDZ4 and PDZ5 show any, even remote, similarities that between PDZ8 and PDZ9 in the INADL/PLCβ4 interface?

Thanks for the question. To compare the inter-PDZ domain packing interface between INAD PDZ45 and INADL PDZ89, we aligned the INAD PDZ45 and INADL PDZ89 structures from the two complexes as shown in Author response image 2. We found several similarities of the two PDZ coupling interface: (i) The secondary structural elements that are involved in the inter-PDZ domain packing for the two PDZ tandems are similar (Author response image 2, with key residues involved in the inter-domain coupling highlighted in blue dots in Author response image 2). (ii) The C-terminal tail extension of both PDZ tandems are involved in the inter-domain coupling. In INAD PDZ45, the C-terminal extension of PDZ5 binds to a surface on PDZ4 and stabilizes the PDZ45 coupling. Although we did not observe clear electron densities of the C-terminal extension of PDZ9 in the INADL/PLCβ4 complex structure, biochemical analysis showed that removal of PDZ9 C-terminal extension weakened the complex interaction by ~5-fold (∆CT in Figure 5I), suggesting that this C-terminal extension is also involved in the PDZ89 coupling. We have added the above points in the revised manuscript.

**Author response image 2. respfig2:** Comparison of the inter-PDZ domain packing of the INAD PDZ45 and INADL PDZ89 tandems in the two complex structures. (**A**) Superposition of INAD PDZ45 (in yellow) and INADL PDZ89 (in green). (**B**) Ribbon combined stick representation showing the domain packing interface of INAD PDZ45. (**C**) Ribbon combined stick representation showing the domain packing interface of INADL PDZ89. (**D**) Multiple sequence alignment of INAD PDZ45 and INADL PDZ89 from *Drosophila* to human by ClusterW and ESpript (espript.ibcp.fr/ESPript/ESPript/). The key residues involved in the PDZ domain coupling are highlighted in blue dots. The key residues of PDZ domains involved in the NORPA/PLCβ4 interaction interface are highlighted by black dots.

4) The authors found that decreasing the pH of the buffer from 7.8 to 5.8 dramatically weakened the binding between INAD and NORPA by ~ 6700 fold (Figure 3E). Can this be rationalized by the structure of the complex?

This point has been covered in our previous work (Liu et al., 2011). Briefly, we demonstrated that a pair of hydrogen bond between the His547 in PDZ4 and Thr669 in PDZ5 tail extension is vital for PDZ45 coupling and lowering pH (a phenomena likely to happen when photoreceptors are activated by light (Huang et al., 2010; Liu et al., 2011) can decouple PDZ45 supramodule due to protonation of the sidechain of His547. In this study, we predicted that if this light activated/pH-mediated inter-PDZ45 decoupling mechanism indeed operates, we should be able to detect a pH-induced binding impairment between PDZ45 and NORPA. Our biochemical data indeed supported this mechanism. The newly added data showing weakened binding between T669E-PDZ45 and NORPA CC-PBM in Author response image 1 also supports this mechanism.

5) The supramodule interaction between CC-PBM and PDZ tandem is supported by solid data and represents a novel functional mechanism of PDZ scaffold proteins. A potential concern is the involvement of this interaction in ipRGC signaling. When the PBM of NORPA is deleted, both activation and deactivation of the fly photo response becomes slow at all light intensities (Shieh et al., 1997). In the case of PLCβ4, in contrast, deletion of PBM did not cause any change in the photo response of ipRGC at high intensity, but slightly increased the deactivation speed of responses to dim light. This effect, if real, indicated a role opposite to the known purpose of assembling GPCR signaling complexes, i.e., to speed up the signaling. Is INADL actually required for the photo response of ipRGC? Have the authors tried to downregulate the expression of INADL, or at least show it colocalizes with PLCβ4, in ipRGC?

We appreciate the reviewer’s comments. We fully agree with the reviewers’ concern that the current evidence is not strong enough to support direct involvement of INADL in the ipRGC signaling. Considering that substantial future work will be required to prove or refute the above point. We feel that, for scientific rigor, it is best to remove the rather preliminarily data on PLCβ4mutant in mice visual system and related discussion from the current manuscript. Instead, we have stated that future studies will be required to test whether the INADL/PLCβ4 interaction identified in this study might be functional in ipRGC signaling or PLCβ4 signaling in other tissues. Accordingly, we have revised the title of our manuscript as “An unexpected PDZ tandem-mediated PLCβ binding critical for photo signal transductions in *Drosophila*” and modified the abstract and text throughout the manuscript.

6) INADL may form signaling complex with PLCβ4 in some cell types, but the occurrence in ipRGC is not convincing at all based on the shown electrophysiological results. Please consider taking out this part of conclusion. If the authors would like to make the manuscript stronger, they could obtain in vivo evidence for the supramodule interaction using fly assays. For instance, the authors could find a point mutation in PDZ4 of INAD that disrupts its NORPA interaction and demonstrate in transgenic flies that this mutation phenocopies PBM-lacking NORPA.

Please see our response to the point above this one.

7) The new structures provide insight of interaction of PDZ domains and their binding proteins. However, the in vivo function of this interaction is lacking, and the conclusion that interactions of PLCβs/PDZ is conserved in ipRGC signaling is not fully supported. It has been previously shown that PDZ1 binds to the C-terminus of NORPA, while PDZ5 binds to an internal region (including in C-terminal coiled-coil domain) (van Huizen et al., 1998), moreover, the crystal structure of the PDZ1 and c-terminal peptide has been determined (Kimple et al., 2001). The authors found that PDZ45 bind strongly to C-terminal regions. Please comment and discuss.

We thank the reviewer’s comments. Indeed, there were reports of INAD PDZ1 and NORPA in the early literatures. In 2001, Kimple et al. determined the crystal structure of INAD PDZ1 in complex with NORPA C-terminal last five amino acid peptide (sequence: TEFCA). Although experiments were carefully performed in these papers, these studies suffered an unanticipated error in the NORPA sequence. Early version of the NORPA sequence deposited in the database contained an error. The second last amino acid residue should be Tyr instead of Cys (i.e., should be EEEAYKTQGKTEFYA instead of EEEAYKTQGKTEFCA; this error has been corrected in the NCBI and uniprot database later). The Tyr residue at the -1 position of NORPA PBM is very important for INAD PDZ45 interaction as shown in Figure 2F, and substitution of Tyr with Ala largely weakened this interaction (Figure 2H). The NORPA sequence error also explained an artificial disulfide bond between INAD PDZ1 and NORPA C-terminal tail in the study by Kimple et al. To provide further evidence, we tested the binding of INAD PDZ1 to the NORPA PBM or CC-PBM with the correct sequence. Fluorescence polarization assay using FITC-labeled NORPA PBM peptide (Author response image 3 below) and FPLC-based assay (Author response image 3) showed no detectable binding between INAD PDZ1 and NORPA PBM or CC-PBM.

**Author response image 3. respfig3:** Binding analysis of INAD PDZ1 to NORPA PBM and NORPA CC-PBM. (**A**) Fluorescence polarization assay showed that no interaction could be detected between NORPA PBM peptide (TQGKTEFYA) and INAD PDZ1. As a control, FITC-labeled NORPA PBM peptide bound to INAD PDZ45 with a *K*_d_ ~ 40 μM. (**B**) FPLC analysis showed INAD PDZ1 does not interact with NORPA CC-PBM.

8) Several key components of fly phototransduction are assembled by 5-PDZ domain protein, INAD, which supporting speed and sensitivity of fly vision. However, although the ipRGC signaling shares high similarity with phototransduction, this signal does not appear to demand the high sensitivity and signal rate. Therefore, it is a question if an INAD-like signal complex exists in the ipRGC signaling. There is no direct evidence that INADL has an in vivo function in the ipRGC signaling as INAD in fly vision. In fly photoreceptor cells, the c-terminal mutations of NORPA caused reduction of NORPA protein in rhabdomeres and severe defects in phototransduction as by slow activation and deactivation. Please comment and discuss.

We thank the reviewer’s comments. We fully agree with the reviewer’s view and removed this part of the conclusion from the manuscript (please also refer to our response to the point #5).

9) Comments on the structures:a) The R_free_ values seem too high for all four structures. Please check the refinements.

Thanks for pointing out this to us. We have further refined all four structures and the re-refined structures have much better R_free_ values (shown in the modified structural statistics table in the revised manuscript). Author response image 4 below shows the statistics of our structures compared to other structures at similar resolutions (generated by the program POLYGON (Urzhumtseva et al., 2009)). The R_free_ values are at the reasonable range now. Also, the structures are with good geometry as indicated by the bond/angle RMSD values as well as the Ramachandran plots. To further show the structural quality, we have generated the phased anomalous map for the two datasets (Se-Met NORPA CC-PBM K884E K898E and Gold-derived PLCβ4 CC-PBM^8KA^/INADL PDZ89) with anomalous signals (Author response image 5). The peaks in the anomalous map (shown in purple) overlap well with the anomalous scatters (i.e. Se atoms of Se-Met and Au, respectively).

**Author response image 4. respfig4:** The statistics of our four structures compared to other structures at similar resolutions generated by the program POLYGON in the PHENIX software suite (Urzhumtseva et al., 2009).

**Author response image 5. respfig5:** Phased anomalous difference electron density map. (**A**) Three representative Se sites (M974, M1003, and M1053) of the Se-Met NORPA CC-PBM K884E K898E structure are shown and the electron density is contoured at a 3.0σ level. (**B**) One representative gold site of the Gold-derived PLCβ4 CC-PBM^8KA^/INADL PDZ89 structure is shown and the electron density is contoured at a 4.0σ level. The gold atom is conjugated to the sulfur of C1047 from PLCβ4 CC-PBM.

b) Please show a difference map of the PDZ45 module after molecular replacement phasing of the NORPA CC^8KA^-PBM/INAD PDZ45 (i.e., before the PDZ45 module was modeled).

In the molecular replacement phasing process, the PDZ45 was placed after the placing of the half of the NORPA CC (which is only 1/4 of the entire structure) instead of the entire NORPA. We could hardly see the electron density of INAD PDZ45. To show the PDZ45 was modeled correctly, we generate an omit map by deleting the PDZ45 module from the final model. Author response image 6 shows the F_o_-F_c_ difference map of the PDZ45 module at different contour levels (2.0σ and 2.5σ levels, respectively). Please note that the PDZ45 module is nearly half of the entire structure, thus the electron density is quite sparse. Nevertheless, we can still find modeled PDZ45 overlaps well with the density.

**Author response image 6. respfig6:** An omit map showing the PDZ45 module. The F_o_-F_c_ density map is generated by deleting the INAD PDZ45 part from the final model and contoured at 2.5σ (**A**) and 2.0σ (**B**), respectively.

c) The authors note that covalently linking INADL PDZ89 with PLCβ4 CC^8KA^-PBM was necessary to obtain diffraction quality crystals of the complex. Please elaborate. Was there no diffraction in the absence of the linker? Also, please provide details of the cross-linking procedure.

Many thanks for the reviewer’s question. After numerous trials, the complex of PLCβ4 CC^8KA^-PBM/INAD PDZ89 without the covalent linker can only be crystallized into tiny needle-like crystals diffracted to 5Å. In order to improve the crystal, we fused INADL PDZ89 to the N-termini of PLCβ4 CC^8KA^-PBM with a 15aa linker (GGGGSGGGGSGGEGS). This has been included in the revised manuscript.

[Editors' note: further revisions were requested prior to acceptance, as described below.]

The manuscript has been improved but there are some remaining issues that need to be addressed before acceptance, as outlined below:The omit map (Author response image 6 in the rebuttal) with the entire PDZ45 module omitted is rather fragmented since it represents half of the entire structure. Please generate a simulated annealing composite omit map instead. Also, this map and representative portions of the final 2mFo-DFc map should be included in a supplementary figure.In the previous version, the authors claimed that the new PDZ-PLC interaction mediated melanopsin signaling in mammalian ipRGC cells despite of some conflicting data. Although this conclusion and the related data have been removed in this revised version, the Introduction and Result sections still emphasize the potential role of this interaction in ipRGC signaling (for example, the Title, second paragraph of the Introduction and subsection “PLCβ4 adopts a similar mode in binding to INADL as NORPA does to INAD”). Please remove these parts, and rather discuss a potential involvement of a PDZL/PLC4 complex in ipRGC signaling at the end of Discussion.The Title should also be softened in order avoid an unproven claim, i.e., that the binding is critical for Drosophila photo signaling.

In response to your decision letter, we have re-revised our manuscript. The following are the changes that we have made during this round of the revision:

1) We have removed the descriptions of ipRGC both in the Introduction and Results section. We have added a very brief discussion on potential link the interaction between INADL/PLCβ4 on ipRGC. We have also softened the Title of the manuscript by revising the Title into “An unexpected INAD PDZ tandem-mediated PLCβ binding in *Drosophila* photo receptors”.

2) Following your suggestion, we have now generated a simulated annealing composite omit map and the final 2mFo-DFc map of the INAD PDZ45/NORPA CC-PBM complex. These two figures are now included as Figure 2—figure supplement 3D,E in the revised manuscript. The quality of both maps is quite reasonable.